# HierLoc: Hyperbolic Entity Embeddings for Hierarchical Visual Geolocation

**Hari Krishna Gadi**[1,2]**, Daniel Matos**[1]**, Hongyi Luo**[1,2]**, Lu Liu**[1]**,**
**Yongliang Wang**[1]**, Yanfeng Zhang**[*,1]**, Liqiu Meng**[2]
[1]Huawei Riemann Lab, Hilbert Research Centre
[2]Chair of Cartography, Technical University of Munich
```
{hari.krishna.gadi1, daniel.matos, hongyi.luo, luliu1}@huawei.com
{wangyongliang775, zhangyanfeng}@huawei.com
{harikrishna.gadi, hongyi.luo, liqiu.meng}@tum.de
```

## Abstract

Visual geolocalization, the task of predicting where an image was taken, remains challenging due to global scale, visual ambiguity, and the inherently hierarchical structure of geography. Existing paradigms rely on either large-scale retrieval, which requires storing a large number of image embeddings, grid-based classifiers that ignore geographic continuity, or generative models that diffuse over space but struggle with fine detail. We introduce an entity-centric formulation of geolocation that replaces image-to-image retrieval with a compact hierarchy of geographic entities embedded in Hyperbolic space. Images are aligned directly to country, region, subregion, and city entities through Geo-Weighted Hyperbolic contrastive learning by directly incorporating haversine distance into the contrastive objective. This hierarchical design enables interpretable predictions and efficient inference with 240k entity embeddings instead of over 5 million image embeddings on the OSV5M benchmark, on which our method establishes a new state-of-the-art performance. Compared to the current methods in the literature, it reduces mean geodesic error by 19.5%, while improving the fine-grained subregion accuracy by 43%. These results demonstrate that geometry-aware hierarchical embeddings provide a scalable and conceptually new alternative for global image geolocation.

## 1 Introduction

Visual geolocalization, inferring where an image was taken from its content alone, is a fundamental challenge with applications in biodiversity monitoring (Van Horn et al., 2021), cultural heritage preservation (DeLozier et al., 2016), news verification (Tahmasebzadeh et al., 2023), and augmented reality (Mercier et al., 2023). However, many real-world images lack geotags in their metadata (Flatow et al., 2015), making automated solutions increasingly important. The task remains difficult due to its scale and ambiguity (Dufour et al., 2025). The search space spans the entire globe; visual patterns such as beaches or skylines recur across continents; language similarities in the street view images span continents; and geographic space itself is structured hierarchically from continents down to cities.

Most existing methods follow one of three paradigms: retrieval-based, classification, and more recently, generative methods. Retrieval-based approaches index millions of image embeddings and return nearest neighbors (Haas et al., 2023), which capture fine-grained similarity but do not scale gracefully and provide limited interpretability. Classification methods (Astruc et al., 2024; Haas et al., 2024) tackle this task by predicting a discrete cell, respecting geography but failing to capture cross-continental visual relationships. Generative models, such as diffusion, can model spatial uncertainty, but underperform retrieval methods at fine scales (Dufour et al., 2025).

We present a geolocation architecture that models the world as a hierarchy of entities such as countries, regions, subregions, and cities, and learns embeddings for each entity instead of indexing

---

*Author to whom correspondence should be addressed to

individual images. Images are aligned to entity embeddings via a contrastive loss weighted by the scaled haversine distance. Conventional retrieval methods scale linearly, requiring $O(N)$ comparisons against millions of images; approximate nearest neighbor methods reduce runtime but still incur large memory and indexing overheads. Our approach instead operates on a fixed, compact set of entity embeddings which scale sub-linearly, enabling faster inference through hierarchical traversal: predictions are resolved coarsely at higher levels and refined only where needed. This allows for beam search over the hierarchy, allowing efficient exploration of plausible paths. This yields scalable inference, interpretable outputs, and even potential for client-side deployment. By reframing geolocation as "image-to-entity alignment" rather than "image-to-image retrieval," the method makes the structure of geography central to the task.

To encode the geographic hierarchy effectively, we represent entities in hyperbolic space. In Euclidean space, hierarchical structures become increasingly compressed as depth grows: the number of entities expands roughly exponentially from country $\rightarrow$ region $\rightarrow$ subregion $\rightarrow$ city, but Euclidean distances grow only linearly. This mismatch causes deeper-level entities to crowd together, reducing discriminative power during inference. Hyperbolic geometry, by contrast, naturally provides exponential volume growth and therefore allocates proportionally more space to represent large branching factors in deep hierarchies (Nickel & Kiela, 2017; Chen et al., 2021). As a result, related entities can remain close, while fine-grained locations can still be well separated, making hyperbolic space a more faithful and expressive embedding space for geographic hierarchies. This effect can be observed visually in the Figure 2 in the Appendix A.2. To our knowledge, this is the first application of Hyperbolic embeddings to represent hierarchical geographic entities for geolocation explicitly. To couple HYperbolic geometry with geographic structure, we introduce a *Geo-Weighted Hyperbolic InfoNCE* (GWH-InfoNCE) loss that weights negative logits using the haversine formula. This objective materially improves fine-scale discrimination while preserving global structure.

We evaluate our method on OSV5M (Astruc et al., 2024), comprising 4.8 million training and 200k test images, and on MediaEval'16 (Larson et al., 2017) with 4.7 million training images. On OSV5M, our approach establishes a new state of the art, yielding consistent gains across all hierarchical levels: country (+8.8%), region (+20.1%), subregion (+43.2%), and city (+16.8%), while reducing mean geodesic error by 19.5% relative to the strongest baselines. We further confirm robustness on IM2GPS (Hays & Efros, 2008), IM2GPS3K (Vo et al., 2017), and YFCC4K (Vo et al., 2017). Beyond these benchmarks, our findings highlight that Hyperbolic embedding spaces provide a principled advantage for multimodal representation learning wherever data exhibit inherent hierarchical structure, with geolocation serving as a particularly suitable testbed. The following are our contributions:

- Reduce search complexity by reformulating geolocation as image-to-entity alignment in Hyperbolic space, cutting the search from millions of images to 240k entities while improving accuracy.

- Demonstrate that Hyperbolic geometry captures multi-scale geographic relationships for hierarchical representation.

- Introduce Geo-Weighted Hyperbolic InfoNCE (GWH-InfoNCE), which incorporates great-circle distance to emphasize geographically proximal negatives.

- Achieve state-of-the-art results on OSV5M across all levels (country +8.8%, subregion +43.2%), validating the effectiveness of geometry-aware learning.

## 2 RELATED WORKS

**Global visual geolocation.** Classical work framed geolocation as image retrieval against large galleries, e.g. IM2GPS (Hays & Efros, 2008), later revisited with stronger deep learning baselines (Vo et al., 2017).Early pioneers like PlaNet (Weyand et al., 2016) and CPlaNet (Seo et al., 2018) used S2 cells to discretize the globe into multi-scale classes. Recent iterations like Where We Are and What We're Looking at Clark et al. (2023) and PIGEON (Haas et al., 2024) have refined this via massive-scale classification and semantic fusion. While recent methods emphasize scalability (SC retrieval (Haas et al., 2023), PIGEON (Haas et al., 2024)) or generative modeling of geodesic uncertainty (Dufour et al., 2025). This trend continues with the work of LocDiff (Wang et al., 2025) with multi-scale latent diffusion. Benchmarks such as MediaEval'16 and OSV5M (Astruc et al., 2024) further

standardized evaluation. The emergence of foundational models also resulted in works such as Geo-Reasonser (Li et al., 2025) and Img2Loc (Zhou et al., 2024), which push the boundary forward. Several hybrid approaches combine retrieval and classification, such as Translocator (Pramanick et al., 2022) and GeoDecoder (Clark et al., 2023).

While existing geo-classification methods use multi-scale cells for partitioning, they often collapse into a single-level output, losing critical hierarchical signals. Our method explicitly preserves and leverages this hierarchical structure during both representation and inference.

**Hyperbolic deep learning and vision.** Hyperbolic spaces such as the Poincaré disk and Lorentz model are well-suited to tree-like structures due to exponential volume growth (Nickel & Kiela, 2017; Ganea et al., 2018). Poincaré embeddings (Nickel & Kiela, 2017) and hyperbolic neural networks (Ganea et al., 2018) established this line of work, with applications in vision showing advantages over Euclidean and spherical embeddings for hierarchical classification (Khrulkov et al., 2020). However, to our knowledge, no prior work has embedded a global geolocation hierarchy (continent → city) in hyperbolic space or evaluated such embeddings on standard geolocation benchmarks. Our approach adapts the Lorentz model for stable training and cross-modal alignment.

**Location encoders.** Compact embeddings of raw geographic coordinates support geo-aware perception, e.g. Space2Vec (Mai et al., 2020), Sphere2Vec (Mai et al., 2023), and GeoCLIP (Cepeda et al., 2023). These methods treat coordinates directly as prediction targets. In contrast, we embed *geographic entities* enriched with multimodal features (image, text, coordinates) into hyperbolic space, yielding interpretable prototypes that unify hierarchical structure with cross-modal signals.

## 3 METHODOLOGY

### 3.1 HYPERBOLIC GEOMETRY

We operate in the Lorentz (hyperboloid) model of Hyperbolic space $\mathbb{H}_K^d$ with constant curvature $-1/K$ (Ganea et al., 2018; Ratcliffe, 2019). All neural operations are performed in the tangent space at the canonical origin $o = (\sqrt{K}, 0, \dots, 0)$, using the exponential and logarithmic maps.

$$\exp_O(v) = \left( R \cosh\left(\tfrac{\|v\|}{R}\right),\ R \sinh\left(\tfrac{\|v\|}{R}\right) \tfrac{v}{\|v\|} \right), \qquad \log_O(x) = \frac{R \operatorname{arcosh}\left(\tfrac{x_0}{R}\right)}{\sqrt{x_0^2 - R^2}} \vec{x}, \qquad (1)$$

$$d_{\mathbb{H}}(x, y) = \operatorname{arcosh}\left( -\tfrac{\langle x, y \rangle_{\mathcal{L}}}{K} \right) \qquad (2)$$

with $R = \sqrt{K}$, and $x = (x_0, \vec{x})$. This decomposition of $x$ is due to the Lorentz model's different treatment of the first component compared to the subsequent ones, see Appendix A.2 for more information. Also, $d_{\mathbb{H}}(x, y)$ denotes the geodesic distance. More details about the general forms are presented in Appendix A.2. Since neural layers rely on vector-space operations (linear maps, bias additions), they are not directly well-defined on $\mathbb{H}_K^d$ (Ganea et al., 2018). Our model therefore performs all such operations in the flat tangent space at the origin: inputs $x \in \mathbb{H}_K^d$ are mapped to $\log_O(x)$, transformed in $\mathbb{R}^d$, and lifted back via $\exp_O(x)$. Fixing the base point to $o$ provides (i) a unique, global reference shared across all entities, (ii) closed-form exp/log maps with efficient implementation, and (iii) stable training without introducing additional learnable base points. This choice is standard in Hyperbolic neural networks and ensures outputs remain valid points on $\mathbb{H}_K^d$. Some libraries (e.g. `geoopt`[1]) parametrize the hyperboloid as $\langle x, x \rangle_{\mathcal{L}} = -k$. Our $K$ corresponds exactly to this $k$, so curvature is $-1/K$ and radius $R = \sqrt{K}$.

### 3.2 CONSTRUCTION OF HIERARCHY AND ENTITIES

We construct a hierarchical tree of geographic entities directly from the training metadata (Algorithm 1 in Appendix A.3). The hierarchy spans four levels: *Country*, *Region*, *Subregion*, and *City*. At each level $h$, we define the entity set $\mathcal{E}_h$ as the collection of unique geographic units observed in the metadata (e.g., ISO2 codes for countries, canonical region names within countries, etc.). Entities

---

[1]https://github.com/geoopt/geoopt

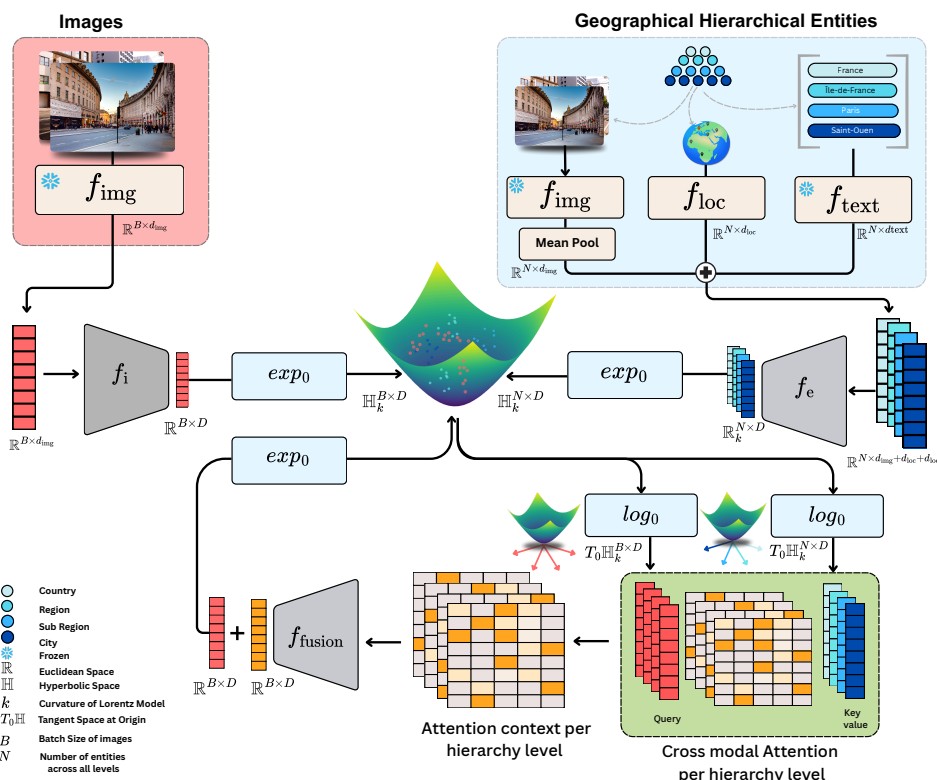

Figure 1: **HierLoc.** Overall architecture. Images are encoded and mapped with $\exp_0$ into the Lorentz model of Hyperbolic space, while entities (countries, regions, subregions, cities) combine image, text, and location features. In the tangent space at the origin, cross-modal attention aligns each image with entities per hierarchy level; the resulting attention outputs are fused and projected back via $\exp_O$. Entity embeddings are not updated with cross-attention context, while image embeddings are updated using the context of cross-modal attention. Training employs our proposed Geo-Weighted Hyperbolic InfoNCE (GWH-InfoNCE), which reweights negatives with the haversine formula between image and negative entity coordinates.

are assigned stable identifiers by concatenating ISO2 codes with sanitized region, subregion, and city tokens. Each training image is then mapped to a tuple of four entities $(e_{\text{country}}, e_{\text{region}}, e_{\text{subregion}}, e_{\text{city}})$. For OSV5M, we use the official quadtree-aligned labels provided with the dataset. For MediaEval'16, where only coordinates are available, we obtain labels through deterministic reverse geocoding with Nominatim[2] and apply canonicalization rules for consistent identifiers (details in Appendix A.3). Each entity $e_i \in \mathcal{E}$ is associated with three multimodal features: an image embedding $\text{Img}_i \in \mathbb{R}^{d\text{img}}$, a text embedding $\text{Text}_i \in \mathbb{R}^{d\text{text}}$, and geographic coordinates $\text{Coords}_i \in \mathbb{R}^2$. The image embedding $\text{Img}_i$ and coordinate features $\text{Coords}_i$ are computed as averages over all training images linked to the entity, using a frozen image encoder $f_{\text{img}}$ (DINOV3 (Siméoni et al., 2025), unless otherwise specified) and their latitude/longitude metadata. The text embedding is derived from the entity name via a pretrained CLIP text encoder (Radford et al., 2021) represented by $f_{\text{text}}$. While averaging may seem like a crude choice, at the entity level it produces stable and discriminative prototypes: the mean embedding captures the dominant visual signal of an entity and is sufficient to distinguish it from other entities in a contrastive learning setup. The role of an entity representation is not to distinguish among all the images assigned to it, but to capture enough shared signal to reliably separate it from other entities at the same hierarchical level. This construction also yields a dramatic compression of the training metadata. Across both datasets, roughly 9.6 million image records are distilled into about 240k entities: 233 countries, 4,946 regions, 29,214 subregions, and 209,894 cities. This reduces the search space from millions of raw images to a compact set of entity prototypes without sacrificing discriminative power. The result is a compact, interpretable,

---

[2]https://github.com/osm-search/Nominatim

and computationally efficient representation of the geographic hierarchy. Low-level implementation details such as sanitization, key construction, and reverse-geocoding policies are deferred to Appendix A.3. Please note that we train two different models seperately please see section 4 for more details.

### 3.3 ENTITY EMBEDDINGS

We fix the tangent space dimension to $d = 128$, so all update vectors lie in $\mathbb{R}^{128}$. Each entity $e_i$ is assigned an *anchor embedding* $A_i \in \mathbb{H}_K^d$, initialized by sampling $\epsilon_i \sim \mathcal{N}(0, \sigma^2 I_d)$ in the tangent space at the origin $T_0 \mathbb{H}_K^d \cong \mathbb{R}^d$, and mapping with $A_i = \exp_O(\epsilon_i)$. This anchor serves as a stable reference point for the entity on the hyperboloid. Each entity is also associated with multimodal features from Section 3.2: an averaged image embedding $\mathrm{Img}_i$, a text embedding $\mathrm{Text}_i$, and normalized coordinates $\mathrm{Coords}_i \in [0,1]^2$. The coordinates are passed into the SphereM+ location encoder (Mai et al., 2023), yielding $\phi_i^{\mathrm{loc}} = f_{\mathrm{loc}}(\mathrm{Coords}_i)$. Together with $\phi_i^{\mathrm{img}} = \mathrm{Img}_i$ and $\phi_i^{\mathrm{text}} = \mathrm{Text}_i$, these features are concatenated and mapped to the tangent space by a three-layer MLP with Dropout and GELU activations (except after the last layer):

$$u_i = \mathrm{MLP}_{\mathrm{ent}}([\phi_i^{\mathrm{loc}} \| \phi_i^{\mathrm{img}} \| \phi_i^{\mathrm{text}}]), \qquad \Delta_i = W_\Delta u_i + b_\Delta \in \mathbb{R}^d.$$

Here $\Delta_i$ is a learnable tangent-space update vector, while $W_\Delta$ and $b_\Delta$ are the weights and biases of a linear projection. The final entity embedding is then obtained by updating the anchor in the tangent space:

$$H_i = \exp_O\big(\log_0(A_i) + \alpha_{\mathrm{node}}\Delta_i\big) \in \mathbb{H}_K^d,$$

where $\alpha_{\mathrm{node}}$ is a learnable scalar controlling update strength. Thus, $A_i$ defines a stable initialization, while $\Delta_i$ injects multimodal evidence to adapt the entity embedding. Entities therefore reside in the 128-dimensional Hyperbolic manifold $\mathbb{H}_K^{128}$, represented in Lorentz coordinates in $\mathbb{R}^{129}$. Distances are computed via the Lorentz inner product in this $(d+1)$-dimensional ambient space.

### 3.4 IMAGE EMBEDDINGS

Each input image is encoded by $f_{\mathrm{img}}$ into a Euclidean feature vector $\phi^{\mathrm{img}} \in \mathbb{R}^{d_{\mathrm{img}}}$. This representation is mapped into the tangent space at the origin by a projection MLP with two linear layers (Dropout and GELU applied after the first layer):

$$u^{\mathrm{img}} = \mathrm{MLP}_{\mathrm{img}}(\phi^{\mathrm{img}}), \qquad \Delta^{\mathrm{img}} = W_{\mathrm{img}} u^{\mathrm{img}} + b_{\mathrm{img}} \in \mathbb{R}^d.$$

We then prepend a zero to $\Delta^{\mathrm{img}}$ to respect the $(d+1)$-dimensional Lorentz structure, and project it to the hyperboloid using the mapping in Eq. 1:

$$Z^{\mathrm{img}} = \exp_O\big(\alpha_{\mathrm{img}} \Delta^{\mathrm{img}}\big) \in \mathbb{H}_K^d,$$

where $\alpha_{\mathrm{img}}$ is a learnable scale. Unlike entities, image embeddings from a frozen backbone are projected to the tangent space at the origin of the Lorentz model with an MLP layer, $f_i$. They are then mapped onto the Lorentz model with $\exp_O$ and later refined through a cross-modal attention module (Section 3.5) with entity embeddings. Both image embeddings $Z^{\mathrm{img}}$ and entity embeddings $H_i$ therefore reside in $\mathbb{H}_K^{128}$, a 128-dimensional Hyperbolic manifold, and are represented in $\mathbb{R}^{129}$, for distance computations we use Eq. 2.

### 3.5 CROSS-MODAL ATTENTION

As shown in Figure 1, cross-modal attention operates in the tangent space at the origin, with images as queries, and entity features as keys/values. All cross-modal interactions are carried out in the tangent space at the origin. For each hierarchy level $\ell \in \{\mathrm{country}, \mathrm{region}, \mathrm{subregion}, \mathrm{city}\}$, entity embeddings $H_j^\ell \in \mathbb{H}_K^d$ are mapped to $h_j^\ell = \log_0(H_j^\ell)$ and each image $Z_{\mathrm{img}}$ to $z_{\mathrm{img}} = \log_0(Z_{\mathrm{img}})$. At level $\ell$, multihead attention is applied with the image as queries, and the entities as keys and values, yielding $\tilde{z}_{\mathrm{img}}^\ell = \mathrm{Attn}_\ell(z_{\mathrm{img}}, \{h_j^\ell\}_j)$. We use a multihead attention block with 8 heads for each level. The four level-wise contexts are concatenated, fused by a small MLP, and added back to the original feature.

$$z_{\mathrm{img}}^\star = z_{\mathrm{img}} + \mathrm{MLP}_{\mathrm{fuse}}([\tilde{z}_{\mathrm{img}}^{\mathrm{country}} \| \tilde{z}_{\mathrm{img}}^{\mathrm{region}} \| \tilde{z}_{\mathrm{img}}^{\mathrm{subregion}} \| \tilde{z}_{\mathrm{img}}^{\mathrm{city}}]),$$

which is then lifted back via $Z_{\mathrm{img}}^\star = \exp_O(z_{\mathrm{img}}^\star) \in \mathbb{H}_K^d$. Only the image stream is updated with attention outputs; entity embeddings remain fixed, an asymmetry that prevents the overfitting of entity embeddings on the training data while still providing hierarchical geographic context.

## 3.6 GWH-INFONCE LOSS

We propose *Geo-Weighted Hyperbolic InfoNCE* (GWH-InfoNCE), a novel contrastive objective that incorporates geographic structure into Hyperbolic alignment. For an image embedding $Z_{\text{img}}^\star \in \mathbb{H}_K^d$, the entity at level $\ell$ provides the positive $H_\ell^+$, while all other entities in that level serve as negatives $\{H_{\ell,k}^-\}_k$. Distances are measured directly on the Lorentz manifold using Eq. 2:

$$d_\ell^+ = d_\mathbb{H}(Z_{\text{img}}^\star, H_\ell^+)^2, \qquad d_{\ell,k}^- = d_\mathbb{H}(Z_{\text{img}}^\star, H_{\ell,k}^-)^2. \tag{3}$$

To emphasize geographical spatial proximity in the embedding space, we reweight each negative according to its great-circle distance $g_{\ell,k}$ from the image location, computed via the haversine formula (Appendix A.4). The per-level loss is

$$\mathcal{L}_\ell = -\log \frac{\exp(-d_\ell^+/\tau)}{\exp(-d_\ell^+/\tau) + \sum_k w_{\ell,k} \exp(-d_{\ell,k}^-/\tau)}, \qquad w_{\ell,k} = 1 + \lambda \exp(-g_{\ell,k}/\sigma). \tag{4}$$

We use Laplace decay for geo-weighting with $\exp(-g_{\ell,k}/\sigma)$, which can also be ablated using other decaying functions such as Gaussian and Inverse kernels. We determine the optimal kernel for weight decay through experiments (see Appendix A.7 for more details). Here $\tau, \lambda, \sigma$ are learnable hyperparameters, with $\tau$ representing the temperature scaling, $\lambda$ and $\sigma$ control the strength of geographic weighting and geographic distance scaling, respectively. The total training objective aggregates across hierarchy levels to minimize,

$$\mathcal{L} = \sum_{\ell \in \mathcal{H}} \beta_\ell \mathcal{L}_\ell,$$

where $\beta_\ell$ trades off supervision across levels, allowing for finer or coarser granularity. We optimize entity and image parameters in Euclidean and Hyperbolic space, respectively (details in Section 4).

## 4 DATASETS AND EXPERIMENTS

### 4.1 DATASETS

We evaluate our method using two large-scale training datasets and several standard benchmarks. Specifically, we train two separate models on the OSV5M and MediaEval'16 datasets respectively. OSV5M (Astruc et al., 2024) contains 4.8 million street-view images for training and 210,000 images for testing; we follow the official split and report results on the test set for direct comparison with published baselines. MediaEval'16 (Larson et al., 2017) provides 4.7 million geo-tagged images, all of which we use for training since no official split is publicly available, and we evaluate on external benchmarks. Specifically, we test a model trained on MediaEval'16 with YFCC4K (Vo et al., 2017), IM2GPS (Hays & Efros, 2008), and IM2GPS3K (Vo et al., 2017), which are widely used standard datasets in visual geolocation.

#### 4.1.1 METRICS

On OSV5M, we follow the official evaluation protocol and report five metrics: classification accuracy at the country, region, subregion, and city levels; the mean geodesic error (computed as the average great-circle distance $\delta$ between predicted and ground-truth coordinates); and the GeoScore, inspired by the GeoGuessr game[3], defined as $5000 \times \exp(-\delta/1492.7)$ (Dufour et al., 2025). The possible range of values for GeoScore is from 0 to 5000. For YFCC4K, IM2GPS, and IM2GPS3K, we report the standard "% @ km" recall statistics used in prior geolocalization work (Haas et al., 2024). This measures the percentage of predictions within fixed distance radii of the ground-truth: 1 km (street-level), 25 km (city-level), 200 km (region-level), 750 km (country-level), and 2500 km (continent-level). We also report the median distance error for these datasets.

#### 4.1.2 EXPERIMENTS

Given a query image and its embedding, we retrieve predictions using a beam search procedure over entity embeddings at each hierarchy level. At each step, candidates are ranked by the Hyperbolic geodesic distance defined in Section 3.1, and the top-$k$ candidates are retained, allowing

---

[3]https://www.geoguessr.com/

Table 1: Geolocation performance comparison on OSV5M with official training and test splits with current baselines. Best results are reported in bold, second-best results are underlined.

| Method | GeoScore ↑ | Dist. (km) ↓ | Classification Accuracy (%) ↑ | | | |
|---|---|---|---|---|---|---|
| | | | Country | Region | Subregion | City |
| SC 0-shot    (Haas et al., 2023) | 2273 | 2854 | 38.4 | 20.8 | 9.9 | 14.8 |
| Regression    (Astruc et al., 2024) | 3028 | 1481 | 56.5 | 16.3 | 1.5 | 0.7 |
| ISNs    (Muller-Budack et al., 2018) | 3331 | 2308 | 66.8 | 39.4 | – | 4.2 |
| Hybrid    (Astruc et al., 2024) | 3361 | 1814 | 68.0 | 39.4 | 10.3 | 5.9 |
| SC Retrieval    (Haas et al., 2023) | 3597 | 1386 | 73.4 | 45.8 | 28.4 | 19.9 |
| RFM $S_2$    (Dufour et al., 2025) | 3767 | 1069 | 76.2 | 44.2 | – | 5.4 |
| LocDiff    (Wang et al., 2025) | - | - | 77.0 | 46.3 | – | 11.0 |
| **HierLoc (VITL-14) (ours)** | 3850 | 1067 | 80.1 | 52.9 | 39.0 | 22.2 |
| **HierLoc (DINOV3) (ours)** | **3963** | **861** | **82.9** | **55.0** | **40.7** | **23.3** |

exploration of multiple plausible locations. The final prediction is selected from the best-scoring beam. This retrieval strategy leverages the hierarchical structure of our entity embeddings, making beam search computationally feasible. We use a beam width of $k = 10$ throughout. The procedure yields classification accuracies at the country, region, subregion, and city levels on OSV5M. At the city level, the coordinates of the predicted entity serve as the image's location estimate, from which we compute mean geodesic error and GeoScore on OSV5M, and distance-based recall and median error on YFCC4K, IM2GPS, and IM2GPS3K. All nearest-neighbor lookups use FAISS FlatIP with a time-coordinate flip, ensuring Lorentz inner products can be ranked efficiently without explicit distance computation (for more details see Appendix A.8). For training, we use AdamW for Euclidean parameters and RiemannianAdam (Bécigneul & Ganea, 2019) for manifold parameters, with gradient clipping for stability. Models are trained with a batch size of $B = 16$ images, a learning rate of $2 \times 10^{-4}$, and run on 6×NVIDIA L40S GPUs for 5 epochs. Each full training run requires approximately 60 hours.

### 4.1.3 RESULTS

Table 1 summarizes the results for the large-scale OSV5M benchmark. HierLoc is trained on OSV5M dataset and tested on OSV5M test set. HierLoc achieves a GeoScore of 3963 and reduces the mean geodesic error to 861 km, a significant improvement over retrieval-based baselines such as SC Retrieval (1386 km) and even the generative RFM $S_2$ model (1069 km). At the same time, HierLoc sets new state-of-the-art classification accuracies across all hierarchy levels, reaching 82.93% at the country level and 23.26% at the city level. These gains highlight the effectiveness of combining Hyperbolic embeddings with beam search retrieval to exploit the hierarchical structure of geographic entities. Since all of the current baselines use the VITL-14 backbone for fair comparison, we also train HierLoc with the VITL-14 backbone and report the results, surpassing the current baselines on all the metrics. This proves our results are not only because of the DINOV3 backbone but also because of our framework. To ensure further fair comparisons with baselines since the model RFM $S_2$ is trained on StreetCLIP (Haas et al., 2023) backbone, we have ablated the choice of backbone and show HierLoc's performance boost does not depend on the choice of the Encoder. We have included these ablations in the tables 4, 5.

Table 2 extends the evaluation to the long-standing IM2GPS, IM2GPS3K, and YFCC4K benchmarks for the model trained on MediaEval'16. On IM2GPS, HierLoc achieves a median error of 21.4 km while maintaining strong recall at large scales (92.4% @ 2500 km). On IM2GPS3K, HierLoc balances fine-grained and coarse performance, cutting the median error nearly in half relative to PIGEON (72.7 km vs. 147.3 km) and yielding +7.1 points improvement at 25 km recall. On YFCC4K, our model lowers the median error to 341.9 km and improves recall at the city (30.2% @ 25 km) and regional (43.3% @ 200 km) levels, demonstrating robustness beyond landmark-centric datasets.

Finally, Table 3 compares HierLoc against the generative RFM $S_2$ models (Dufour et al., 2025) on YFCC4K. Despite being trained on only 4.7M images, HierLoc outperforms the 1M-iteration RFM $S_2$ variant and matches the mean geodesic error of the much larger 10M-iteration model trained on 48M images (2058 km). Although slightly behind in GeoScore and continent-scale recall, HierLoc achieves competitive city and region level performance, highlighting the efficiency of our approach

Table 2: Comparison of HierLoc model with DinoV3 Bbackbone trained on MediaEval'16 data to baselines on benchmark datasets. Median distance error (km, lower is better) and recall (% within radius, higher is better) best results are in bold, and the second-best results are underlined.

| | IM2GPS (Hays & Efros, 2008) | | | | | |
|---|---|---|---|---|---|---|
| Method | Median (km) ↓ | 1 km↑ | 25 km↑ | 200 km↑ | 750 km↑ | 2500 km↑ |
| PlaNet    (Weyand et al., 2016) | > 200 | 8.4 | 24.5 | 37.6 | 53.6 | 71.3 |
| CPlaNet    (Seo et al., 2018) | > 200 | 16.5 | 37.1 | 46.4 | 62.0 | 78.5 |
| ISNs (M,f*,S3)    (Muller-Budack et al., 2018) | > 25 | 16.9 | 43.0 | 51.9 | 66.7 | 80.2 |
| Translocator    (Pramanick et al., 2022) | > 25 | 19.9 | 48.1 | 64.6 | 75.6 | 86.7 |
| GeoDecoder    (Clark et al., 2023) | ∼ 25 | **22.1** | 50.2 | **69.0** | 80.0 | 89.1 |
| PIGEON    (Haas et al., 2024) | 70.5 | 14.8 | 40.9 | 63.3 | 82.3 | 91.1 |
| GeoReasoner    (Li et al., 2025) | - | 13.0 | 44.0 | - | **86.0** | - |
| **HierLoc (ours)** | **21.4** | 10.5 | **51.9** | 67.5 | 83.1 | **92.4** |
| | IM2GPS3K (Vo et al., 2017) | | | | | |
| PlaNet    (Weyand et al., 2016) | > 750 | 8.5 | 24.8 | 34.3 | 48.4 | 64.6 |
| CPlaNet    (Seo et al., 2018) | > 750 | 10.2 | 26.5 | 34.6 | 48.6 | 64.6 |
| ISNs (M,f*,S3)    (Muller-Budack et al., 2018) | ∼ 750 | 10.5 | 28.0 | 36.6 | 49.7 | 66.0 |
| Translocator    (Pramanick et al., 2022) | > 200 | 11.8 | 31.1 | 46.7 | 58.9 | 80.1 |
| GeoCLIP    (Cepeda et al., 2023) | - | 14.1 | 34.5 | 50.6 | 69.7 | 83.8 |
| GeoDecoder    (Clark et al., 2023) | > 200 | 12.8 | 33.5 | 45.9 | 61.0 | 76.1 |
| PIGEON    (Haas et al., 2024) | 147.3 | 11.3 | 36.7 | 53.8 | 72.4 | **85.3** |
| Img2Loc    (Zhou et al., 2024) | - | **17.1** | **45.1** | 57.8 | 72.9 | 84.6 |
| GeoReasoner    (Li et al., 2025) | - | 10.0 | 38.0 | - | **83.0** | - |
| **HierLoc (ours)** | 73.4 | 11.3 | 43.8 | **58.4** | 74.1 | 85.1 |
| | YFCC4K (Vo et al., 2017) | | | | | |
| PlaNet    (Weyand et al., 2016) | > 750 | 5.6 | 14.3 | 22.2 | 36.4 | 55.8 |
| CPlaNet    (Seo et al., 2018) | > 750 | 7.9 | 14.8 | 21.9 | 36.4 | 55.5 |
| ISNs (M,f*,S3)    (Muller-Budack et al., 2018) | > 750 | 6.7 | 16.5 | 24.2 | 37.5 | 54.9 |
| Translocator    (Pramanick et al., 2022) | > 750 | 8.4 | 18.6 | 27.0 | 41.1 | 60.4 |
| GeoDecoder    (Clark et al., 2023) | ∼ 750 | 10.3 | 24.4 | 33.9 | 50.0 | 68.7 |
| PIGEON    (Haas et al., 2024) | 383.0 | 10.4 | 23.7 | 40.6 | **62.2** | **77.7** |
| Img2Loc    (Zhou et al., 2024) | - | **14.1** | 29.5 | 41.4 | 59.2 | 76.8 |
| **HierLoc (ours)** | 341.9 | 8.4 | **30.2** | **43.3** | 61.7 | 75.8 |

Table 3: Comparison of HierLoc with RFM $S_2$, a generative model on YFCC4K. RFM $S_2$ is trained on 48 million YFCC dataset; in contrast, HierLoc is trained on MediaEval'16 with 4.7 million images (10x fewer images). Median distance error (km, lower is better) and recall (% within radius, higher is better). Best results are in bold, second-best results are underlined.

| | YFCC4K (Vo et al., 2017) | | | | | |
|---|---|---|---|---|---|---|
| Method | GeoScore ↑ | Mean Distance (km) ↓ | 25 km↑ | 200 km↑ | 750 km↑ | 2500 km↑ |
| RFM $S_2$    (Dufour et al., 2025) | 2889 | 2461 | 23.7 | 36.4 | 54.5 | 73.6 |
| RFM$_{10M}$ $S_2$    (Dufour et al., 2025) | **3210** | **2058** | **33.5** | **45.3** | 61.1 | **77.7** |
| **HierLoc (ours)** | 3189 | **2058** | 30.2 | 43.3 | **61.7** | 75.9 |

relative to generative models trained on 10 times more data. To further isolate the affect of DinoV3 backbone choice of HierLoc, we provide experiments in the Ablations section 4.2. In summary, across OSV5M and standard benchmarks, HierLoc consistently reduces geolocation error relative to previous baselines, setting new state-of-the-art results, especially on OSV5M. These results validate our design choices of hyperbolic entity embeddings, multimodal fusion, and beam search retrieval as a scalable alternative to current geolocation models.

## 4.2 ABLATIONS

Table 4 reports OSV5M results for three different visual backbones (DINOv3, StreetCLIP, ViT-L/14). All encoders produce highly consistent rankings across the hierarchical accuracy levels, and their absolute performance remains stable, with each backbone outperforming all non-HierLoc baselines. This indicates that the gains do not originate from a particular choice of vision model but from the hierarchical hyperbolic design itself. To also isolate the cross-dataset hierarchy construction, we construct the hierarchies only using OSV5M dataset and train a model on OSV5M with StreetCLIP

Table 4: Encoder ablation of models trained and tested on OSV5M. We compare the effect of different vision backbones on the HierLoc framework.

| Backbone | GeoScore ↑ | Dist. (km) ↓ | Classification Accuracy (%) ↑ | | | |
|---|---|---|---|---|---|---|
| | | | Country | Region | Subregion | City |
| HierLoc (DINOv3) | 3963 | 861 | 82.9 | 55.0 | 40.7 | 23.3 |
| HierLoc (StreetCLIP) | 3862 | 1051 | 80.3 | 53.1 | 39.2 | 22.5 |
| HierLoc (ViT-L/14) | 3850 | 1067 | 80.1 | 52.9 | 39.0 | 22.2 |

Table 5: Encoder ablation of models trained on MP16 and tested on YFCC4K.

| Method | GeoScore ↑ | Mean Distance (km) ↓ | 25 km↑ | 200 km↑ | 750 km↑ | 2500 km↑ |
|---|---|---|---|---|---|---|
| HierLoc (DINOv2) | 3106 | 2211 | 28.2 | 42.2 | 59.8 | 74.0 |
| HierLoc (DINOv3) | 3189 | 2058 | 30.2 | 43.3 | 61.7 | 75.9 |

Table 6: Ablation study on the choice of embedding space in HierLoc, evaluated on the OSV5M dataset. Results are reported in terms of GeoScore, mean localization error, and hierarchical classification accuracy.

| Method | GeoScore ↑ | Mean Dist (km) ↓ | Classification Accuracy (%) ↑ | | | |
|---|---|---|---|---|---|---|
| | | | Country | Region | Subregion | City |
| HierLoc (Euclidean) | 3865 | 968 | 81.0 | 51.5 | 37.5 | 21.1 |
| HierLoc (Spherical) | 3364 | 1258 | 75.2 | 31.3 | 15.9 | 4.3 |
| **HierLoc (Hyperbolic)** | **3963** | **861** | **82.9** | **55.0** | **40.7** | **23.3** |

Table 7: Ablation study of HierLoc on OSV5M, demonstrating the importance of GWH-InfoNCE loss and cross attention for fine-grained localization.

| Method | GeoScore ↑ | Mean Dist (km) ↓ | Classification Accuracy (%) ↑ | | | |
|---|---|---|---|---|---|---|
| | | | Country | Region | Subregion | City |
| DINOV3 zero shot | 2962 | 1999 | 58.7 | 25.8 | 17.1 | 9.6 |
| HierLoc w/ InfoNCE | 3840 | 949 | 80.5 | 49.9 | 35.7 | 19.4 |
| HierLoc w/o attention | 2904 | 1366 | 71.5 | 35.6 | 23.5 | 11.4 |
| HierLoc w/o squared distance | 3752 | 1043 | 79.5 | 47.3 | 33.1 | 17.5 |
| HierLoc w/o text and location | 3890 | 1029 | 81.8 | 52.1 | 37.9 | 21.1 |
| **HierLoc (full)** | **3963** | **861** | **82.9** | **55.0** | **40.7** | **23.3** |

Table 8: Comparison of inference strategies on OSV5M (top-1 accuracy). Flat search ignores hierarchy, while beam search enforces path-consistency. Beam width $k=10$ achieves the best accuracy–efficiency trade-off.

| Method | Country | Region | Subregion | City |
|---|---|---|---|---|
| Flat per-level (no hierarchy) | 79.6 | 50.8 | 39.4 | 22.1 |
| Hierarchical (beam=1) | 79.4 | 48.9 | 36.4 | 21.9 |
| Hierarchical (beam=10) | **82.9** | **55.0** | **40.7** | **23.3** |

backbone. The results in the table 4 for StreetCLIP show that cross-dataset hierarchy construction does not have any influence with the performance of HierLoc. Table 5 presents the corresponding analysis on MP16, comparing HierLoc using DINOv2 and DINOv3. There is a drop in performance with DinoV2 but not significant enough to outperform RFM $S_2$ model, further reinforcing that HierLoc's improvements generalize across encoders and datasets.

Table 6 reports on ablation studies in OSV5M, isolating the independent impact of the main components of HierLoc. Replacing the Lorentz model of HierLoc architecture with Euclidean embedding space increases the mean geodesic error to 968 km and lowers accuracy across all hierarchy levels, confirming the advantage of Hyperbolic space for the geolocation task. Moreover, the Spherical

embedding space performs worse than both the Euclidean and Hyperbolic embedding spaces. This can be attributed to higher distance distortions in the Spherical manifold. We further validate this finding on the YFCC4K, IM2GPS, and IM2GPS3K benchmarks (Appendix, Table 9), where Hyperbolic embeddings consistently reduce median error and improve recall compared to Euclidean space.

Table 7 reports the performance of DINOV3 zero shot, without any training, by finding the nearest neighbors from its image embeddings to entities that have been initialized with only mean image embeddings. Substituting our Geo-Weighted Hyperbolic InfoNCE with the standard InfoNCE objective also degrades performance (949 km), particularly at the region and subregion levels, showing that reweighting negatives by geographic distance provides more effective supervision. Removing cross-modal attention between images and entity prototypes leads to the largest error (1366 km), highlighting that hierarchical context and cross-modal attention are critical for fine-grained localization. Finally, replacing the squared distance in Eq. 3 without the square of the distance also shows a worse performance at all levels.

The ablation of removing the text and location modalities shown in the table 7 does reduce the performance a little across all levels, but it is not significant, and the image signal is the most important modality for this task. But the combination of all three modalities results in the best performance. Furthermore, other ablations such as role of mean image embeddings, hyper parameter sensitivity, curvature choice for the Lorentz model and the choice of weight decaying function for the geo-weights in the loss are reported in the Appendix A.5. Further analysis of the computational efficiency of our framework against retrieval and generative methods is also reported in the Appendix A.8, which quantifies the improvements over existing retrieval methods. We perform an ablation study to evaluate the impact of the hierarchical structure by comparing our approach against a flat retrieval baseline (Table 8). We further analyze the effect of beam search width on localization performance. Figure 5 illustrates the beam search inference process on a geographic map, demonstrating how candidate entities are identified and refined across different hierarchical levels using a beam width of 10.

## 5 DISCUSSION

Our experiments demonstrate that HierLoc provides a principled and scalable solution to visual geolocation. By reformulating the task from image-to-image retrieval into image-to-entity alignment in Hyperbolic space, HierLoc consistently outperforms prior methods across large-scale benchmarks. The improvements are most pronounced at fine-grained levels (subregion and city), where modeling hierarchical structure together with geographically weighted contrastive learning delivers significant gains. Moreover, HierLoc remains competitive with recent generative approaches trained on an order of magnitude more data (e.g.,$RFM_{10M}$ $S_2$ ), highlighting the efficiency of our formulation. Beyond accuracy, HierLoc offers several key advantages. First, predictions are interpretable: images are aligned to explicit geographic entities, enabling structured error analysis and clearer insights into model behavior. Second, inference is computationally efficient, as the number of entities is vastly smaller than the number of training images required for large-scale retrieval. Third, the framework naturally integrates multimodal signals such as text and coordinates improving performance. While Hyperbolic embeddings have been studied extensively, our contributions lie in extending them to planet-scale geolocation, introducing a geo-aware loss in Hyperbolic space, and reformulating retrieval as learnable entity representation learning.

Several limitations remain, 1 km localization performance of our method is inherently bounded by a discretization limit resulting from our selection of 240k fixed entities, which lacks dense street-level granularity. We acknowledge this as a fundamental trade-off: our hierarchical representation prioritizes structural and regional interpretability over sub-kilometer accuracy. Our evaluation in Appendix A.9 further demonstrates difficulties in performance generalization between regions with less number of training samples.

Looking ahead, the broader promise of HierLoc lies in its generality. Any task with hierarchically structured data such as taxonomies in biodiversity, linguistic families, or knowledge graphs could benefit from the same principles of Hyperbolic entity embeddings, multimodal fusion, and structured retrieval. Visual geolocation thus serves as a challenging and high-impact testbed, but the underlying methodology extends beyond it.

## 6 ETHICS STATEMENT

This work relies exclusively on publicly available datasets for visual geolocation, including OSV5M, MediaEval'16, YFCC4K, IM2GPS, and IM2GPS3K, which do not contain personally identifiable information beyond image content already released for research purposes. No additional human subjects were involved, and no private or sensitive data were collected. We acknowledge that geolocation technologies may pose privacy and security risks if misapplied, for example in surveillance, tracking individuals, or identifying sensitive locations. Our work is intended solely for scientific benchmarking and methodological advancement in large-scale representation learning. All datasets are cited from their original sources and used under their research licenses. We encourage future research to consider fairness and bias issues, particularly regarding underrepresented geographic regions, and to assess societal impacts of deploying such models.

## 7 REPRODUCIBILITY STATEMENT

We have taken several steps to ensure reproducibility of our results. All datasets used are publicly available and are described in Section 4 and Appendix A.3, with preprocessing steps (e.g., entity construction, reverse geocoding) detailed in Algorithm 1 and Appendix A.3. Model architecture, hyperparameters, and training schedules are described in Sections 3 and 4, with additional ablations (e.g., curvature sensitivity, kernel functions) reported in Appendix A.5. Evaluation protocols strictly follow prior work and official benchmarks. We also detail how FAISS can be leveraged with Hyperbolic nearest neighbor search in the Appendix A.8.

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

## A  APPENDIX

### A.1  LLM USAGE

During the preparation of this paper, large language models (LLMs) were used as assistive tools for writing and polishing text. Specifically, they were employed to improve the clarity, grammar, and style of certain sections, and to suggest alternative phrasings in accordance with the ICLR author guidelines. LLMs were not involved in research ideation, implementation, experimental design, or analysis of results. All technical content, including methodology, experiments, and conclusions, were designed, implemented, and validated by the authors. The authors take full responsibility for the correctness and originality of the content.

### A.2  PRELIMINARIES ON HYPERBOLIC GEOMETRY

We use the Lorentz (hyperboloid) model of $d$-dimensional Hyperbolic space (Ratcliffe, 2019). This is a Riemannian manifold $\mathbb{H}_K^d$ with constant negative sectional curvature (-1/K) (Ganea et al., 2018).

$$\mathbb{H}_K^d = \big\{\, x \in \mathbb{R}^{d+1} : \ \langle x, x \rangle_{\mathcal{L}} = -K, \ x_0 > 0 \,\big\}, \qquad K > 0,$$

with Minkowski bilinear form

$$\langle x, y \rangle_{\mathcal{L}} \; = \; -x_0 y_0 \; + \; \sum_{i=1}^{d} x_i y_i.$$

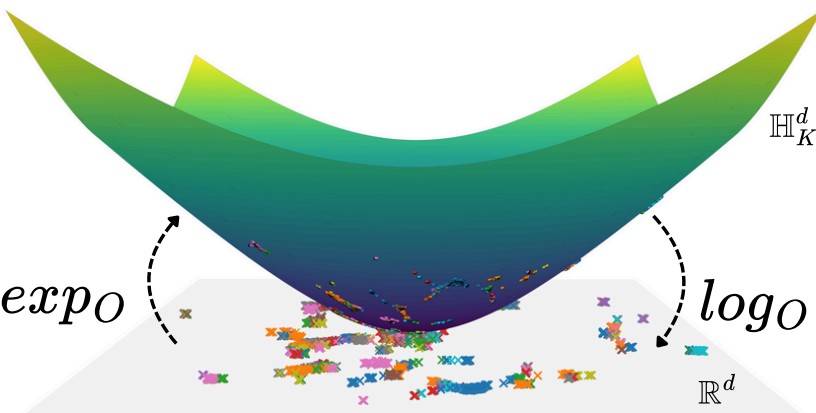

Figure 2: Illustration of $\exp_O$ and $\log_O$ projection functions projecting points from Tangent space to Hyperbolic space and vice versa.

The geodesic distance between $x, y \in \mathbb{H}_K^d$ is

$$d_{\mathbb{H}}(x, y) \;=\; \text{arcosh}\left( -\frac{\langle x,y \rangle_{\mathcal{L}}}{K} \right). \tag{5}$$

It is often convenient to write the radius as $R = \sqrt{K}$, with curvature $K = -1/R^2$.

For any base point $p \in \mathbb{H}_K^d$, the tangent space is $T_p\mathbb{H}_K^d = \{v \in \mathbb{R}^{d+1} : \langle p, v \rangle_{\mathcal{L}} = 0\}$. Let $\|v\|_{\mathcal{L}} := \sqrt{\langle v, v \rangle_{\mathcal{L}}}$ denote the (spacelike) Lorentz norm on $T_p\mathbb{H}_K^d$. With $R = \sqrt{K}$, the *exponential* and *logarithmic* maps at $p$ are

$$\exp_p(v) \;=\; \cosh\left(\tfrac{\|v\|_{\mathcal{L}}}{R}\right) p \;+\; R \sinh\left(\tfrac{\|v\|_{\mathcal{L}}}{R}\right) \frac{v}{\|v\|_{\mathcal{L}}}, \qquad v \in T_p\mathbb{H}_K^d, \tag{6}$$

$$\log_p(x) \;=\; \frac{d_{\mathbb{H}}(p, x)}{\sinh\left(\frac{d_{\mathbb{H}}(p,x)}{R}\right)} \left( x - \cosh\left(\tfrac{d_{\mathbb{H}}(p,x)}{R}\right) p \right), \qquad x \in \mathbb{H}_K^d, \tag{7}$$

with the continuous extensions $\exp_p(O) = p$ and $\log_p(p) = O$. Let the canonical origin be $o = (R, 0, \ldots, 0) = (\sqrt{K}, 0, \ldots, 0) \in \mathbb{H}_K^d$, so $T_0\mathbb{H}_K^d = \{(0, v_1, \ldots, v_d)\} \cong \mathbb{R}^d$. Replacing $p = o$ in Eqs. 6–7 gives

$$\exp_O(v) \;=\; \left( R\cosh\left(\tfrac{\|v\|}{R}\right), \; R\sinh\left(\tfrac{\|v\|}{R}\right) \frac{v}{\|v\|} \right), \qquad v \in \mathbb{R}^d, \tag{8}$$

$$\log_O(x) \;=\; \frac{R \, \text{arcosh}\left(\frac{x_0}{R}\right)}{\sqrt{x_0^2 - R^2}} \, \vec{x}, \qquad x = (x_0, \vec{x}) \in \mathbb{H}_K^d, \tag{9}$$

where $\|v\|$ is the Euclidean norm of $v$, and $\vec{x} \in \mathbb{R}^d$ are the "spacelike" coordinates of $x$. Note that $\log_O(x)$ is not simply the Euclidean projection of $\vec{x}$; the prefactor ensures exact agreement with the Riemannian logarithm. Figure 2 illustratively shows the process of projecting points from $\mathbb{R}^d$ to $\mathbb{H}_K^d$ using $\exp_O$ function, conversely, the process of projecting points from $\mathbb{H}_K^d$ to $\mathbb{R}^d$ using $\log_0$ function.

### A.3 FURTHER DETAILS ON CONSTRUCTION OF ENTITIES

We construct the entity hierarchy directly from the metadata of the train splits of OSV5M and MediaEval'16 datasets. For each dataset, we first resolve schema columns (country, region, subregion, city, latitude, longitude, image embedding) in a format-agnostic way. Each row is then normalized: the country field is mapped to ISO2 code, canonical name; missing labels country, region, subregion, and city labels are filled by reverse geocoding the coordinates with Nominatim[4] in the

---

[4]https://github.com/osm-search/Nominatim

case of MediaEval'16; and all region, subregion, and city names are sanitized to ensure consistent identifiers.

The hierarchy is built incrementally: for every row, we traverse from country to region, subregion, and city, creating nodes as needed. Each node accumulates a count of images, the sum of coordinates, and the sum of image embeddings, which are returned from a frozen image encoder backbone such as DINOV3 or VITL-14. After all rows are processed, entity features are finalized by averaging accumulated values: the mean image embedding, the mean latitude/longitude coordinates, and a text embedding computed from the entity name (via a frozen pretrained text encoder such as CLIP Text Encoder).

Finally, the tree is serialized. The construction runs in linear time in the number of rows and requires memory proportional to the number of unique entities.

---

**Algorithm 1:** BuildHierarchyFromMetadata($\text{dataset}$)

---

**Input:** Training metadata of a dataset $\text{dataset}$
**Output:** Hierarchy tree $T$
1   **Initialize:** $T \leftarrow \emptyset$ (hierarchy starts at countries);
2   **foreach** *record $(c, r, s, ci, \phi, \lambda, v)$ in metadata* **do**
3      Map country $\rightarrow$ (ISO2, canonical name);
4      **if** $\text{dataset} = \texttt{MediaEval'16}$ *and labels missing* **then**
5         $(c, r, s, ci) \leftarrow$ ReverseGeocode$(\phi, \lambda)$ via Nominatim;
6      Sanitize $(r, s, ci)$ to canonical tokens;
7      **foreach** *level $\in$ [country, region, subregion, city]* **do**
8         $u \leftarrow$ CreateOrGetNode($T$, level, identifiers);
9         Update counts and coordinate sums at $u$;
10        Accumulate image embedding $v$ at $u$ (from frozen encoder);

11   **foreach** *node $n \in T$* **do**
12      $\text{Img}_n \leftarrow$ mean of image embeddings;
13      $\text{Coords}_n \leftarrow$ mean of lat/lon;
14      $\text{Text}_n \leftarrow f_{\text{text}}(\text{Name}(n))$ (frozen text encoder);
15   Serialize $T$ to JSON;
16   **Complexity:** $O(N)$ over records $N$; memory $\propto$ number of unique entities;

---

### A.4   HAVERSINE DISTANCE

We compute the geographic distances $g_{\ell,k}$ used in Eq. 4. For two locations $(\varphi_1, \lambda_1)$ and $(\varphi_2, \lambda_2)$ in radians (latitude, longitude), define

$$a = \sin^2\left(\frac{\varphi_2 - \varphi_1}{2}\right) + \cos\varphi_1 \cos\varphi_2 \sin^2\left(\frac{\lambda_2 - \lambda_1}{2}\right).$$

The haversine formula yields the central angle (great-circle distance in radians) as

$$g_{\text{full}} = 2 \arcsin\left(\sqrt{a}\right) = 2 \arctan 2\left(\sqrt{a}, \sqrt{1-a}\right).$$

In our implementation, we omit the constant factor of 2 and work with

$$g = \arcsin\left(\sqrt{a}\right),$$

since both the factor 2 and the Earth's mean radius $R = 6371\,\text{km}$ simply rescale distances without changing their relative ordering or gradient directions. These constants are absorbed into the bandwidth parameter $\sigma$ of our loss. For reporting physical distances, we reintroduce them as

$$d = 2Rg \quad \text{(in kilometers)}.$$

### A.5   FURTHER ABLATIONS

**Manifold sensitivity**   In addition to the OSV5M ablation (Table 6), we also compare Euclidean and Hyperbolic variants of HierLoc on three widely used recall-at-km benchmarks: YFCC4K, IM2GPS, and IM2GPS3K (Table 9). The Hyperbolic manifold consistently outperforms Euclidean space across datasets, reducing median localization error by 23–45% and improving recall at city- and region-level thresholds (25–200 km). This confirms that the advantages of Hyperbolic embeddings are robust and not specific to OSV5M.

Table 9: Manifold sensitivity on recall-at-km benchmarks (YFCC4K, IM2GPS, IM2GPS3K). Same backbone, loss, and training schedule; only the embedding manifold differs. Hyperbolic space consistently reduces median localization error and improves recall across scales.

| Dataset | Median (km) ↓ | 1 km ↑ | 25 km ↑ | 200 km ↑ | 750 km ↑ | 2500 km ↑ |
|---|---|---|---|---|---|---|
| **YFCC4K (Vo et al., 2017)** | | | | | | |
| HierLoc (Euclidean) | 445.3 | 7.0 | 25.9 | 39.7 | 58.2 | 73.4 |
| HierLoc (Hyperbolic) | **341.9** | **8.4** | **30.2** | **43.3** | **61.7** | **75.8** |
| **IM2GPS (Hays & Efros, 2008)** | | | | | | |
| HierLoc (Euclidean) | 47.3 | 8.4 | 45.9 | 64.9 | 81.8 | 91.1 |
| HierLoc (Hyperbolic) | **21.4** | **10.5** | **51.9** | **67.5** | **83.1** | **92.4** |
| **IM2GPS3K (Vo et al., 2017)** | | | | | | |
| HierLoc (Euclidean) | 121.6 | 10.2 | 41.2 | 55.1 | 71.6 | 83.3 |
| HierLoc (Hyperbolic) | **73.4** | **11.3** | **43.8** | **58.4** | **74.1** | **85.1** |

Table 10: Ablation study of removing mean image embeddings for each hierarchy level sequentially

| Ablation (Removed Mean Embeddings) | Country (%) | Region (%) | Sub-region (%) | City (%) |
|---|---|---|---|---|
| None | 82.93 | 55.03 | 40.68 | 23.26 |
| Country | 81.59 | 54.27 | 39.97 | 22.63 |
| Country & Region | 79.96 | 50.40 | 36.61 | 20.59 |
| Country & Region & Sub-region | 77.12 | 44.34 | 30.76 | 16.32 |
| Country & Region & Sub-region & City | 70.46 | 30.00 | 14.06 | 3.58 |

Table 11: Ablation study of HierLoc on OSV5M subset of 100k training set and 10k test set for search of the best curvature, $K$ for Lorentz model.

| Curvature ($K$) | Mean Dist (km) ↓ | Classification Accuracy (%) ↑ | | | |
|---|---|---|---|---|---|
| | | Country | Region | Subregion | City |
| 0.25 | 1719 | 64.7 | 28.8 | 17.7 | 8.36 |
| 0.50 | 1603 | 67.4 | 31.6 | 19.9 | 9.51 |
| 0.75 | 1534 | 69.1 | 33.5 | 21.1 | 10.0 |
| **0.80** | **1462** | **69.2** | **33.7** | **22.3** | **11.0** |
| 1.00 | 1687 | 67.4 | 30.2 | 19.3 | 9.1 |

**Mean Image Embeddings Sensitivity** To quantify the contribution of mean image embeddings across spatial scales, we progressively remove them in a coarse-to-fine order (country → region → subregion → city). Table 10 shows that removing country-level means yields only a minor accuracy drop, indicating that coarse aggregates provide limited discriminative signal. The impact grows as we remove means from finer levels, with the largest decline at the city level where local visual context is most informative. Importantly, the model remains stable and does not collapse even when all mean embeddings are removed. This demonstrates that HierLoc is not dependent on any single level's mean representation; rather, its performance stems from the hierarchical architecture and the multi-level integration of visual cues.

Table 11 shows the search of the best curvature, $K$, of the Lorentz model on the OSV5M subset of 100k training images and 10k test images. Owing to the size of the dataset, we perform this search on a smaller subset. Through the experiments, we find that curvature 0.8 best fits the OSV5M dataset, given that it is the primary dataset we are focusing on. We keep the same curvature for both models on the OSV5M dataset and also the MediaEval'16 dataset.

### A.6 HYPERPARAMETER SENSITIVITY ANALYSES

We extend our analysis to two further hyperparameters of HierLoc's training objective: the temperature $\tau$ and the geographic weighting coefficient $\lambda$ used in the GWH-InfoNCE loss. All experi-

ments are conducted on a 100k/10k OSV5M split. Table 12 reports the performance of HierLoc for $\tau \in \{0.07, 0.10, 0.15, 0.30\}$. The model exhibits stable behavior across a broad range, with $\tau = 0.1$ yielding the best overall accuracy

Table 12: Temperature sensitivity of the GWH-InfoNCE loss on OSV5M (100k/10k split).

| $\tau$ | Country (%) | Region (%) | Sub-region (%) | City (%) |
|---|---|---|---|---|
| 0.07 | 68.72 | 32.76 | 20.62 | 10.31 |
| 0.10 | **69.20** | **33.70** | **22.30** | **11.00** |
| 0.15 | 69.00 | 32.50 | 20.50 | 9.80 |
| 0.30 | 67.80 | 32.30 | 20.00 | 9.67 |

Table 13 varies the geographic weighting coefficient $\lambda$ in the GWH-InfoNCE loss. The model reaches peak performance at $\lambda = 1.0$, but accuracy degrades smoothly as $\lambda$ moves away from this value, demonstrating that the method is not overly sensitive to the choice of weighting strength. Across both hyperparameters, HierLoc maintains stable performance and does not exhibit collapse within a wide range of settings. The smooth variation in accuracy further confirms that the method is robust to moderate deviations from the default $\tau = 0.1$ and $\lambda = 1.0$, and does not rely on finely tuned hyperparameter values.

Table 13: Sensitivity to the geographic weighting parameter $\lambda$ in GWH-InfoNCE.

| $\lambda$ | Country (%) | Region (%) | Sub-region (%) | City (%) |
|---|---|---|---|---|
| 0.0 | 63.90 | 27.80 | 16.20 | 7.30 |
| 0.5 | 67.80 | 32.00 | 20.20 | 9.60 |
| 1.0 | **69.20** | **33.70** | **22.30** | **11.00** |
| 2.0 | 65.00 | 29.50 | 17.70 | 8.20 |

### A.7 GEOGRAPHIC WEIGHTING KERNELS.

Our loss defined in the Eq. 4 can incorporate a family of distance-dependent kernels $k(d)$ that transform geographic distance $d$ (in kilometers) into a similarity weight. Let $d$ denote the great-circle distance in kilometers between the image and a negative entity, $\sigma > 0$ a scale parameter, $\lambda > 0$ a weight strength, and $p > 0$ an exponent. We support three kernel families:

- **Laplace kernel (default):** Exponential decay in distance:

$$k_{\text{Laplace}}(d) = \exp\left(-\tfrac{d}{\sigma}\right),$$

- **Gaussian kernel:** Squared-distance decay, producing a narrower band of influence:

$$k_{\text{Gauss}}(d) = \exp\left(-\left(\tfrac{d}{\sigma}\right)^2\right),$$

- **Inverse kernel:** Power-law decay, yielding long-range tails:

$$k_{\text{Inv}}(d) = \left(1 + \tfrac{d}{\sigma}\right)^{-p}.$$

The final geographic weight for a negative sample, which upweights negatives that are geographically close to the positive is

$$w(d) = 1 + \lambda\, k(d).$$

We observe in Table 14 that the choice of kernel has a measurable effect on geolocation performance. The Gaussian kernel, which decays very rapidly with squared distance, yields reasonable accuracy at coarse levels (country and region) but underperforms at finer scales, since moderately close negatives receive almost no weight and thus provide little contrastive pressure. The Inverse kernel, with its heavy-tailed power-law form, performs slightly better at coarse levels but fails to emphasize fine-scale discrimination, as distant negatives retain substantial weight and dominate the denominator.

Table 14: Ablation study of HierLoc on OSV5M subset of 100k training set and 10k test set for search of the best geoweighting decay kernel for GWH-InfoNCE

| kernel ($k(d)$) | Mean Dist (km) ↓ | Classification Accuracy (%) ↑ | | | |
|---|---|---|---|---|---|
| | | Country | Region | Subregion | City |
| Gauss | 1525 | 69.7 | 33.7 | 21.1 | 10.1 |
| Inverse | 1529 | **69.8** | **34.1** | 21.7 | 10.3 |
| **Laplace** | **1462** | 69.2 | 33.7 | **22.3** | **11.0** |

In contrast, the Laplace kernel achieves the best trade-off: it decays exponentially with distance, preserving emphasis on geographically nearby negatives without entirely discarding moderately distant ones. This balance leads to superior performance at finer levels (subregion and city), where distinguishing between visually similar but geographically close entities is most critical. Moreover, Laplace also reduces the mean geodesic error, showing that its weighting improves localization precision overall.

We therefore adopt the Laplace kernel as the default geo-weighting strategy for GWH-InfoNCE, as it provides the most effective compromise between local discrimination and global robustness in the hierarchical geolocation setting.

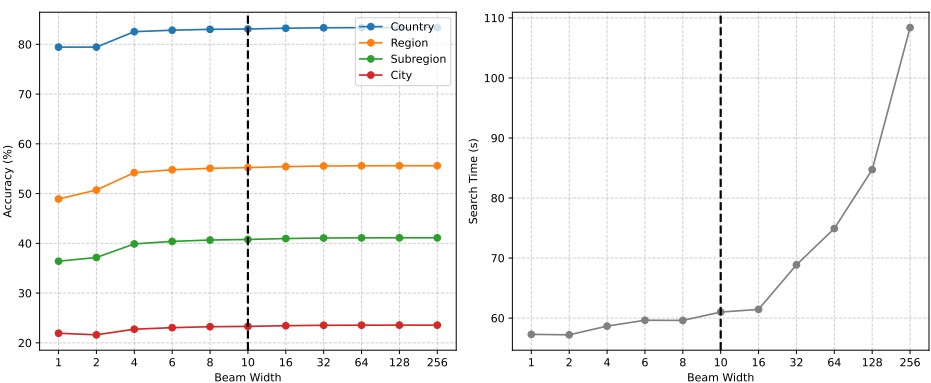

Figure 3: Comparison of accuracy and search time tradeoff with different beam widths.

## A.8 COMPUTATIONAL AND STORAGE EFFICIENCY

Modern retrieval systems typically use libraries such as *FAISS*[5] for fast nearest-neighbor search, and this can be leveraged for Hyperbolic distances in the Lorentz model as well. Figure 4 highlights the efficiency advantages of HierLoc over standard image-based retrieval. While most research benchmarks rely on datasets with millions of images, real-world platforms can be orders of magnitude larger: for example, Mapillary reports more than 2 billion street-level images (Mapillary Team, 2024). At such scales, SC Retrieval, which grows linearly with database size, becomes computationally and storage prohibitive. By contrast, HierLoc scales sublinearly because the number of geographic entities expands far more slowly than the number of raw images. This structural difference translates into significant savings: HierLoc reduces wall-clock inference time by more than $10\times$, achieves over $20\times$ lower storage requirements, and cuts FLOPs per query by two orders of magnitude, all while sustaining higher throughput. Although beam search introduces a slight overhead relative to flat entity search due to sequential parent–child expansion, it yields higher accuracy at all levels with negligible additional cost. Overall, HierLoc achieves a favorable trade-off between scalability and precision, enabling efficient billion-scale deployment that is infeasible with standard retrieval pipelines.

---

[5]https://github.com/facebookresearch/faiss

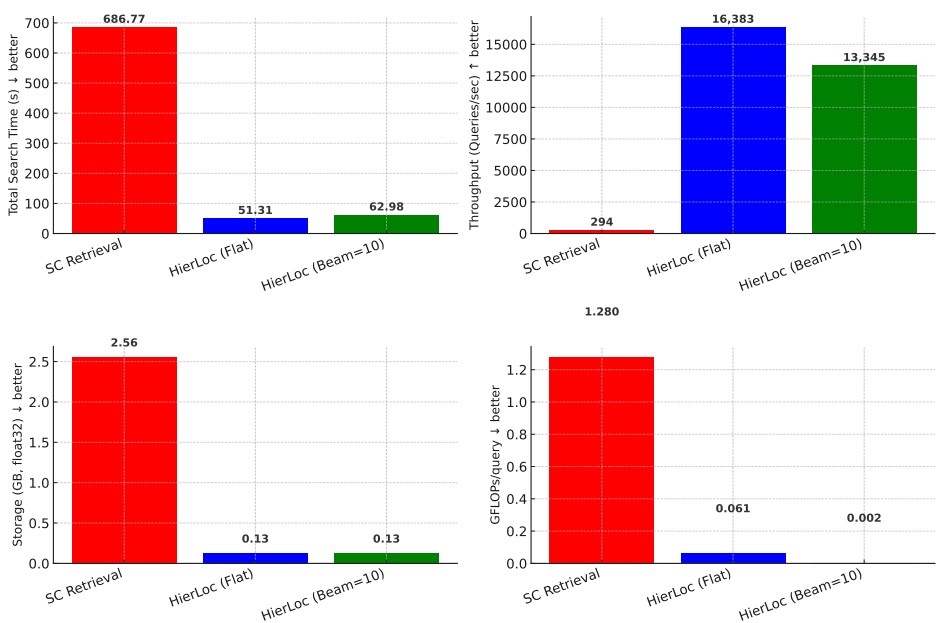

Figure 4: Comparison of computational efficiency across methods. We report wall-clock search time, throughput (queries per second), storage footprint, and FLOPs per query. Arrows indicate whether lower or higher values are better.

**FAISS for Hyperbolic (Lorentz) search and beam expansion.** In the Lorentz model, geodesic distance and the Lorentz inner product are monotone-equivalent:

$$\cosh\big(d_{\mathbb{H}}(x,y)/R\big) \;=\; -\tfrac{1}{K}\,\langle x,y\rangle_L, \qquad \langle x,y\rangle_L \;=\; -x_0 y_0 + \sum_{i=1}^{d} x_i y_i,\; R=\sqrt{K}.$$

Thus minimizing $d_{\mathbb{H}}$ is equivalent to maximizing $\langle\cdot,\cdot\rangle_L$. We convert Lorentz MIPS into standard Euclidean inner-product search supported by FAISS via a one-line linear map on the *query* only: for $Z = (z_0, \mathbf{z})$ define $\tilde{Z} = (-z_0, \mathbf{z})$. Then $\tilde{Z}^\top H = \langle Z, H\rangle_L$, so a FAISS `FlatIP` (or `IVF-FlatIP`/`HNSW-IP`) index over entity embeddings returns the same ranking as sorting by increasing Hyperbolic distance, without evaluating `arcosh` at search time. We do *not* $\ell_2$-normalize Lorentz vectors (normalization would distort the geometry). *Beam-search integration.* We build one FAISS IP index per hierarchy level (country→city) in Lorentz coordinates and: (i) seed the beam at the top level with the top-$k$ entities from a single FAISS query using the time-coordinate flip; (ii) for each deeper level, query the corresponding FAISS index and *parent-filter* candidates so only children of the previous-level beam are retained; (iii) accumulate a path score (e.g., $1 - \mathrm{IP}$ or, for exact scoring of a small shortlist, the true $d_{\mathbb{H}}$) and prune to the fixed beam width. This hybrid "FAISS-guided, parent-constrained" expansion amortizes most work into a few high-throughput IP calls while enforcing path consistency. On GPU, `GpuIndexFlatIP` provides the best throughput; approximate variants (`IVF`/`HNSW`) can be enabled at larger scales with no change to the sign-flip trick. We batch queries, reuse indices across batches, and maintain per-level indices to avoid rebuilds during inference.

To directly assess inference-time efficiency against other diffusion based methods, we compare HierLoc against the strongest non-retrieval baseline, RFM$S_2$, under identical hardware conditions. Table 15 reports the latency breakdown. HierLoc achieves substantially lower forward-pass time and adds only a negligible retrieval cost due to its compact hierarchical search, resulting in an overall $6.56\times$ speedup relative to RFM$S_2$. This demonstrates that the hierarchical entity-based formulation not only improves accuracy, but also yields a highly efficient inference pipeline suitable for large-scale deployment.

Table 15: Inference latency comparison on the same GPU. HierLoc provides a $6.56\times$ speedup over RFM$S_2$.

| Model | Inference (ms) | Retrieval (ms) | Total (ms) | Speedup |
|---|---|---|---|---|
| RFM S2 | 14.17 | – | 14.17 | $1\times$ |
| HierLoc (ours) | 2.09 | 0.075 | 2.16 | $\mathbf{6.56\times}$ |

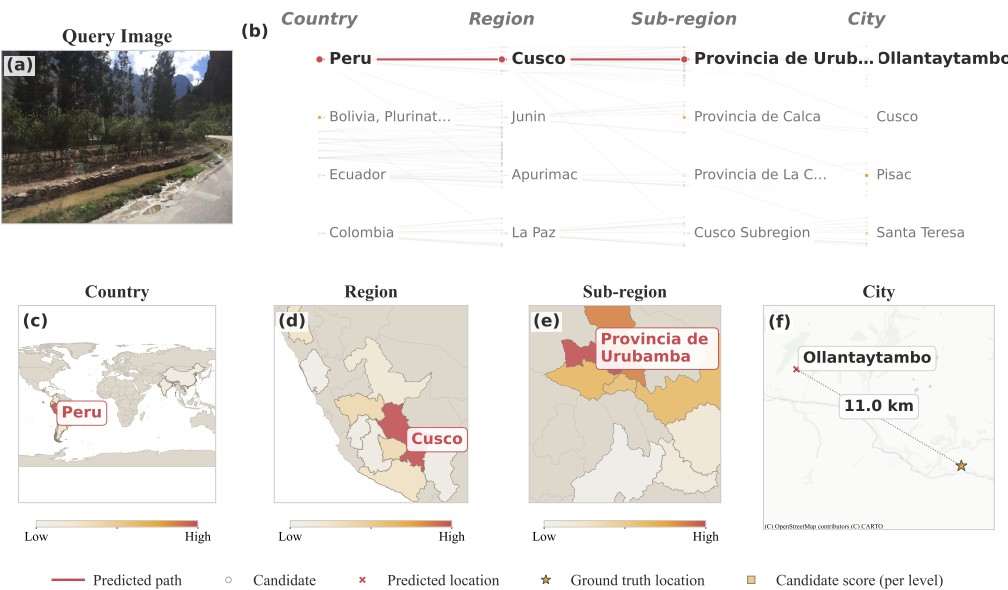

Figure 5: Hierarchical beam search visualization for visual geolocation. Given a query street-view image (a), HierLoc performs a coarse-to-fine geographic search through four hierarchical levels (b). At each level, the model scores candidate regions and selects the highest-scoring entity (red path). Panels (c)–(e) show the candidate regions at each level as a choropleth, where color intensity indicates the model's confidence score normalized within that level. Panel (f) shows the final city-level prediction on an OpenStreetMap basemap, with the predicted location (red cross) and the ground truth image coordinate (gold star).

## A.9 QUALITATIVE ANALYSIS

To provide interpretable insight into the inference behavior of HierLoc, we visualize the full hierarchical beam search trace alongside its geographic realization across the four entity levels in the figure 5. Panel (b) displays the beam search tree, where each column corresponds to a hierarchy level (Country, Region, Sub-region, City) and each node represents a candidate geographic entity scored by computing the squared hyperbolic distance between the query image embedding and the entity embedding in the Lorentz manifold. The predicted path, shown in red, traces the highest-scoring entity selected at each level. Pruned candidates appear as faint background nodes, illustrating the breadth of the search space that the beam width of k=10 efficiently narrows at each step. Panels (c) through (f) project this search onto geographic maps with progressively increasing zoom, mirroring the coarse-to-fine structure of the hierarchy. At the country level (c), all 10 candidate countries are rendered on a world map with choropleth shading proportional to their normalized scores. At the region level (d), the view restricts to regions within the selected country, and similarly for sub-regions (e) within the selected region. This geographic drill-down directly reflects the computational savings of hierarchical beam search, which reduces the candidate space from 240k entities to a tractable subset at each level. The city-level panel (f) overlays the final predicted location and ground truth coordinate on an OpenStreetMap basemap, with a dotted line and distance annotation quantifying the localization error. In this example, HierLoc correctly identifies Peru, narrows to the Cusco re-

gion, localizes to Provincia de Urubamba, and predicts Ollantaytambo within 11.0 km of the true coordinate, demonstrating the model's ability to perform structured geographic reasoning through the learned hyperbolic entity hierarchy.

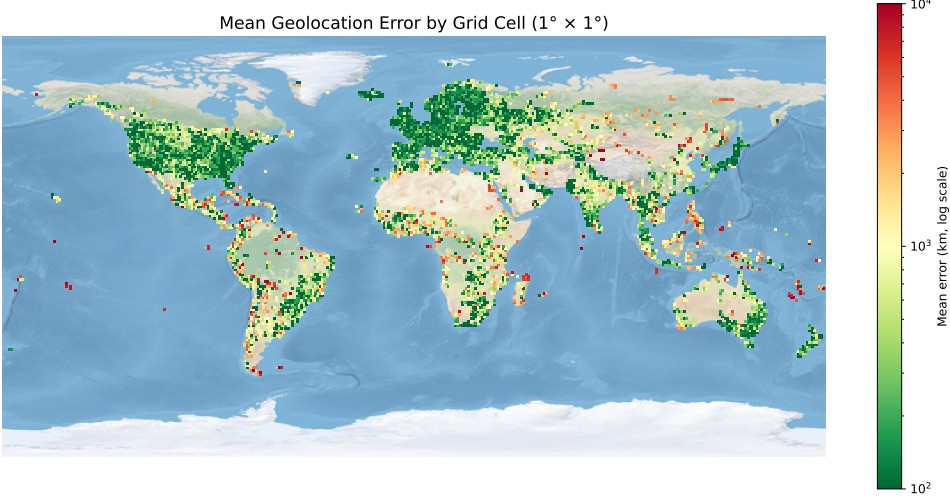

Figure 6: Mean geographic error distribution of HierLoc on the OSV5M dataset.

Figure 6 shows the global distribution of localization errors on OSV5M. Unlike smaller benchmarks such as YFCC4K, IM2GPS, or IM2GPS3K, OSV5M provides a geographically diverse test set with over 200k images, enabling systematic analysis at global scale. Urban regions in Europe and North America exhibit relatively low errors due to dense training coverage, while sparsely imaged regions such as inland Asia and central Africa show higher mean errors, reflecting persistent data imbalance. Because OSV5M is more geographically balanced and diverse than prior datasets (Astruc et al., 2024), performance on this benchmark offers a stronger measure of robustness and generalization. Figure 7 demonstrates that Hyperbolic embeddings, trained solely from image–location pairs, capture meaningful geographic, cultural, and linguistic relationships. Clusters emerge without explicit supervision, reflecting both geographic proximity and cultural ties. Notably, island nations form a distinct peripheral cluster, suggesting that their embedding geometry encodes the same structural differences reflected in their error statistics.

Taken together, these qualitative analyses reveal three systematic error modes: (i) data imbalance, where underrepresented regions (e.g., inland Asia, central Africa) yield higher error; (ii) cultural and linguistic clustering, where embeddings align geographically distant but historically linked nations; and (iii) recurring visual ambiguity in island environments, where beaches and coastlines lack distinctive geographic context and drive long-tail errors. Understanding these modes highlights both the strengths and the remaining challenges for scalable global geolocation.

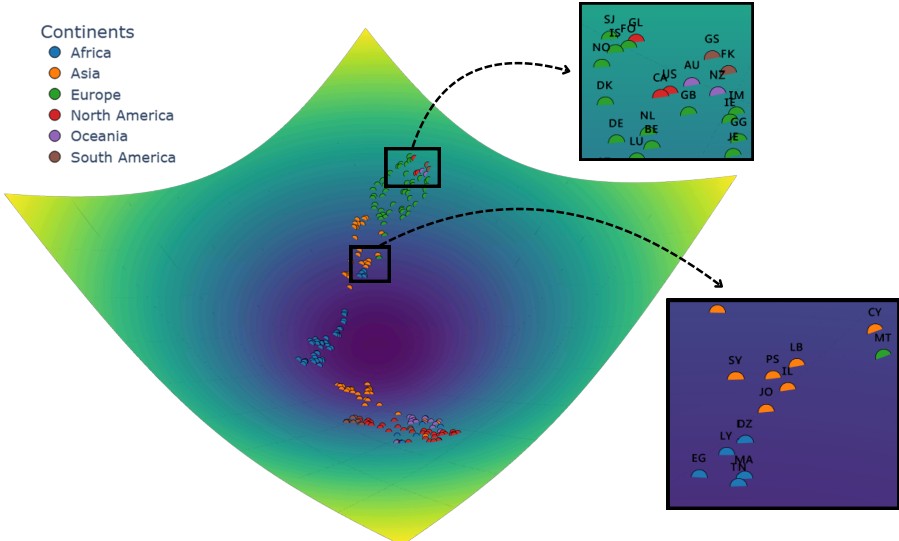

Figure 7: UMAP projection of country embeddings learned in Hyperbolic space. Colors denote continents, with insets highlighting emergent clusters. Geographically distant but culturally linked nations (e.g., Australia, New Zealand, UK, US, Canada) cluster together, while regions with shared history (e.g., the Mediterranean basin) form coherent cross-continental groups. Island nations consistently appear at the periphery of the embedding space.

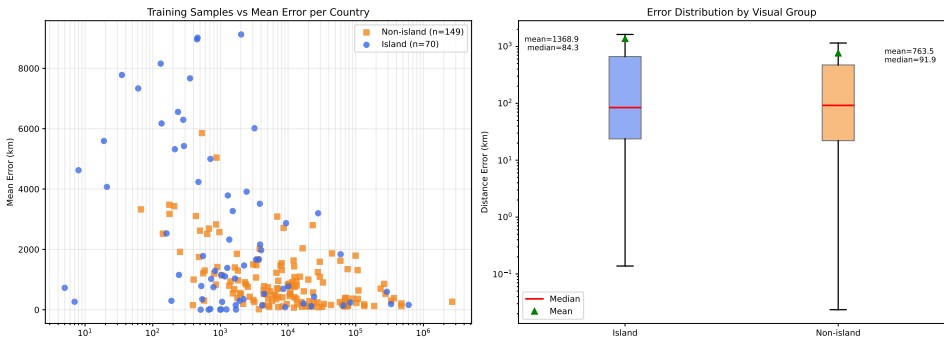

Figure 8: Comparison of geolocation performance across island and non-island territories. **Left:** Scatter plot of mean error versus training samples per country. Both groups benefit from more samples, but islands (blue) show higher variance and extreme outliers. **Right:** Error distributions by group. Although island territories achieve a comparable median error to non-islands (84.3 km vs. 91.9 km), their mean error is nearly twice as high (1368 km vs. 763 km), indicating a heavier long-tail. This reflects the difficulty of disambiguating visually repetitive environments such as coastlines and beaches, where recurring patterns provide weak geographic cues. These challenges are consistent with the peripheral island clusters observed in Figure 7.

