# OpenReview forum: "HierLoc: Hyperbolic Entity Embeddings for Hierarchical Visual Geolocation"
_ICLR.cc/2026/Conference — ICLR 2026 Poster_

### Official Review · Reviewer_7XjS · 2025-10-28

**Soundness:** 2
**Presentation:** 3
**Contribution:** 2
**Rating:** 4
**Confidence:** 3

**Summary:**

The paper trains on two datasets (OSV5M and MediaEval’16) but does not include an ablation showing how each dataset affects performance.
Previous geolocation works (e.g., PIGEON, RFM S, GeoDecoder) trained and evaluated using only one dataset, ensuring fair comparison.

Because HierLoc uses multiple datasets—OSV5M for one benchmark and MediaEval’16 for others—the reported gains may partly come from additional training data diversity, not just from the proposed hyperbolic embedding method.
Without a controlled experiment (e.g., training only on OSV5M and testing cross-dataset), it is unclear whether improvements are architectural or data-driven.

**Strengths:**

The paper is clearly written, with a logical flow.
The figures effectively illustrate the hierarchical process.
Equations are well-defined and consistent with hyperbolic geometry notation.
Introduces the first explicit use of hyperbolic embeddings for visual geolocation, representing geographic hierarchies as hyperbolic manifolds.
The Geo-Weighted Hyperbolic InfoNCE (GWH-InfoNCE) loss incorporating haversine distance is novel and well-justified.
Reformulating geolocation as image-to-entity alignment instead of image-to-image retrieval is conceptually new.

**Weaknesses:**

1. Some hyperparameters are found empirically, more justification or sensitivity plots would be valuable.

2. The paper employs two large-scale datasets (OSV5M and MediaEval’16) for training but does not provide ablations isolating their effects. Prior works (e.g., PIGEON, GeoDecoder, RFM) train and evaluate using a single dataset, ensuring fair comparison. In contrast, HierLoc benefits from additional data diversity, which may contribute to improved cross-dataset generalization. Without an ablation where HierLoc is trained solely on OSV5M or MediaEval’16 and evaluated on the same benchmarks, it is unclear how much of the reported gains are due to the proposed method versus the increased data volume.

I recommend adding an ablation or at least clarifying which datasets were used for each benchmark evaluation, and ensuring fair comparison with prior work under identical training data conditions.

3. The field of visual geolocation is niche within ICLR’s broader ML audience; the paper’s impact may depend on perceived relevance beyond geolocation (e.g., general multimodal hierarchy learning).

**Questions:**

None

---

> ### Author Response · Authors · 2025-11-20
>
> Dear Reviewer,
>
> Thank you for your thoughtful comments and for highlighting important points regarding dataset usage and hyperparameter justification.
>
> **Training Datasets and Fairness of Comparison**
>
> The confusion seems to come from Section 3.2, where entity construction across datasets was described. We apologize for the ambiguity.
>
> To clarify:
>
> **We train two completely separate models**, following standard practice in prior work:
>   - Model A is trained *only on OSV5M* and evaluated on OSV5M.
>   - Model B is trained *only on MediaEval’16 (MP16)* and evaluated on Im2GPS, Im2GPS3K, and YFCC4K.
>
> This mirrors common practice in PIGEON, RFM, GeoDecoder, and GeoCLIP.
> To ensure fairness, we additionally include:
>
> - Ablations where entities are constructed solely from OSV5M or solely from MP16
> - Backbone isolations (DINOv2 vs. DINOv3 vs. StreetCLIP)
>
> as suggested by Reviewer k8zx. Please, see tables 4 & 5 in the revised paper. These isolate dataset influence, entity hierarchy influence, and backbone influence, demonstrating that HierLoc’s gains stem from its **architecture**, not training on bigger dataset.
>
> On OSV5M, the largest open-source test set (210k images)—HierLoc achieves clear SOTA under identical training conditions.
> On Im2GPS, Im2GPS3K, and YFCC4K, HierLoc delivers competitive results while being trained on **4.7M images**, which is **10× fewer** than models like RFM (trained on 48M YFCC images), ensuring fair comparison.
>
> **Hyperparameter Sensitivity**
>
> We already provide ablations on curvature, GWH-InfoNCE weighting kernel, and beam width. To further address your concern, we now include sensitivity analyses for:
>
> - Temperature (τ), with default value 0.1
> - Geographic weighting λ, with default value 1.0
>
> evaluated on a 100k train /10k test OSV5M subset split.
>
> **Rebuttal Table 1. Temperature Sensitivity**
>
> | τ | Country (%) | Region (%) | Sub-region (%) | City (%) |
> |------|------------------|----------------|---------------------|--------------|
> | 0.07 | 68.72 | 32.76 | 20.62 | 10.31 |
> | 0.10 | 69.20 | 33.70 | 22.30 | 11.00 |
> | 0.15 | 69.00 | 32.50 | 20.50 | 9.80 |
> | 0.30 | 67.80 | 32.30 | 20.00 | 9.67 |
>
> **Rebuttal Table 2. Lambda Sensitivity**
>
> | λ | Country (%) | Region (%) | Sub-region (%) | City (%) |
> |------|------------------|----------------|---------------------|--------------|
> | 0.0  | 63.9 | 27.8 | 16.2 | 7.3 |
> | 0.5  | 67.8 | 32.0 | 20.2 | 9.6 |
> | 1.0  | 69.2 | 33.7 | 22.3 | 11.0 |
> | 2.0  | 65.0 | 29.5 | 17.7 | 8.2 |
>
> These results show that HierLoc is robust to the choice of τ and λ; performance varies smoothly and the model does not collapse across a broad range.
>
> **Relevance to the ML / ICLR Community**
>
> Visual geolocation remains a core challenge for multimodal reasoning. Even state-of-the-art vision-language models struggle to infer location reliably, especially at global scale. In the broader pursuit of spatially grounded AI and AGI, our contributions are meaningful:
>
> 1. **Hierarchical multimodal representation learning**, enabling coarse-to-fine spatial reasoning.
> 2. **Hyperbolic entity embeddings at scale**, valuable beyond geolocation for hierarchical classification, knowledge graphs, and structured retrieval.
>
> We appreciate your question about broader impact and have clarified this in the revised paper.
>
> Thank you again for your constructive feedback. We hope these dataset clarifications, controlled experiments, and sensitivity analyses address your concerns.

---

> ### Comment · Reviewer_7XjS · 2025-11-26
> **X**
>
> Thank you for your explanation, especially for addressing the second question.
>
> However, I **personally** have limited interest in the research topic explored in this paper, as the motivation is not clearly articulated and the work does not appear to target a real application.
>
> To clarify, I am not referring specifically to your paper, but rather to most of the studies in this research direction.
>
> Regarding the discussion on spatial understanding for AI and AGI, I believe it is somewhat inconsistent with the viewpoint you presented in your response to Reviewer ZDzY.
>
> From that perspective, the method should be more closely aligned with approaches such as GeoReasoner.

---

> > ### Author Response · Authors · 2025-11-27
> >
> > Dear Reviewer,
> >
> > Thank you for confirming the additional experiments and the clarification regarding weakness 2.
> >
> > We also appreciate your honesty in sharing your personal level of interest in this research direction. For the benefit of readers and the Area Chair, we would like to briefly clarify the real-world motivations of image geolocation as documented in the literature at major conferences.
> >
> > _________________________________________
> >
> > - **Real-world applications highlighted in top-tier papers**
> >
> > PIGEON [1] (section 5) cites applications in autonomous driving, navigation systems, educational tools, open-source intelligence, and investigative journalism.
> >
> > LocDiff [2] (Section 1) describes applications in trajectory synthesis, building segmentation, and large-scale geolocation tasks.
> >
> > RFM [3] (section 1) documents applications in archaeology and cultural heritage (artifact cataloging and interpretation), and in forensic analysis and journalism, where recovering missing GPS data supports crime-scene reconstruction and image verification.
> >
> > GeoReasoner [4] (Section 1) highlights applications in urban planning, social-science research, and street-level navigation.
> >
> > Together, these publications also show the broad real-world relevance of the research and reflect sustained interest across the community and multiple top-tier venues.
> > ______________________________________________________
> >
> > - **Context on Spatial Reasoning and HierLoc Complementarity**
> >
> >
> > We respectfully clarify that there is no inconsistency. "Spatial understanding" in AI requires a spectrum of approaches, ranging from massive reasoning engines to efficient retrieval systems.
> >
> > Efficiency & Privacy: Methods like GeoReasoner [4] rely on LVLMs (Qwen-VL) with billions of parameters, often requiring cloud inference that introduces latency and privacy risks.
> >
> > HierLoc's Role: In contrast, HierLoc provides robust global localization with only 140M parameters. This efficiency enables on-device execution, crucial for privacy-sensitive applications (e.g., autonomous agents, personal photos) where uploading data to a cloud is not feasible.
> >
> > Thus, HierLoc contributes to the broader goal of spatially intelligent AI by solving the scalability and deployment challenges (without compromising on performance) that massive reasoning models currently face.
> >
> > We thank you again for raising these important points and hope these clarifications provide helpful additional context.
> >
> > ___________________________
> >
> > **References**
> >
> > 1. Lukas Haas, Michal Skreta, Silas Alberti, Chelsea Finn; Proceedings of the IEEE/CVF Conference on Computer Vision and Pattern Recognition (CVPR), 2024, pp. 12893-12902.
> >
> > 2. Wang, Zhangyu et al. “LocDiff: Identifying Locations on Earth by Diffusing in the Hilbert Space.” (2025) (accepted to NeurIPS 2025).
> >
> > 3. Nicolas Dufour, Vicky Kalogeiton, David Picard, Loic Landrieu; Proceedings of the IEEE/CVF Conference on Computer Vision and Pattern Recognition (CVPR), 2025, pp. 23016-23026.
> >
> > 4. Ling Li, Yu Ye, Bingchuan Jiang, and Wei Zeng. 2024. GeoReasoner: Geo-localization with reasoning in street views using a large vision-language model. In Proceedings of the 41st International Conference on Machine Learning (ICML'24), Vol. 235. JMLR.org, Article 1176, 29222–29233.

---

> > ### Comment · Reviewer_k8zX · 2025-11-27
> >
> > I would like to respectfully push back against the comment that "The field of visual geolocation is niche within ICLR’s broader ML audience." This characterization contradicts the current trajectory of top-tier machine learning research.
> >
> > Visual geolocation is, in fact, seeing a surge in interest at major generalist ML conferences. For example, NeurIPS 2025 proceedings include multiple prominent works in this domain:
> >
> > - GeoRanker: Distance-Aware Ranking for Worldwide Image Geolocalization
> >
> > - GRE Suite: Geo-localization Inference via Fine-Tuned Vision-Language Models
> >
> > - LocDiff: Identifying Locations on Earth by Diffusing in the Hilbert Space
> >
> > Furthermore, this field has a long-standing history in sister conferences like CVPR (e.g., Revisiting IM2GPS, PIGEON). Crucially for the ICLR community, geolocation is increasingly viewed as a robust proxy for benchmarking VLM reasoning capabilities and representation learning, placing it squarely within the scope of this conference.
> >
> > As a general comment, the reviewers shows a lot of despise towards a whole field. This is not an attitude of someone that can claim to represent the "ICLR community". The reviewers should refrain from expressing their personal preferences, we don't care about it.
> >
> > Reviewing criteria should be based on scientific validity, novelty, and soundness, not on a reviewer's personal interest in the specific sub-field. Gatekeeping a rapidly growing field as "niche" is not a valid ground for rejection. I would even argue that niche subjects might be the ones that have the most potential.  I hope the AC will weigh the technical contributions of this work over subjective comments regarding the topic's popularity.

---

### Official Review · Reviewer_ZDzY · 2025-10-29

**Soundness:** 3
**Presentation:** 3
**Contribution:** 3
**Rating:** 4
**Confidence:** 4

**Summary:**

The paper presents hierloc, a new framework for worldwide image geolocalization that reformulates the task as aligning images to a compact hierarchy of geographic entities, embedded in hyperbolic spaces. Rather than image-to-image retrieval, hierloc computes and aligns image embeddings to entity prototypes via contrastive learning, directly incorporating haversine distance to weight negatives (GWH-InfoNCE). This allows efficient hierarchical beam search inference, interpretability, and strong performance. Empirical results on the OSV5M benchmark and others demonstrate improved mean geodesic error and classificationn accuracies relative to prior methods, while reducing storage and computation requirements.

**Strengths:**

1. The paper is clearly presented, and the reader can smoothly follow the authors’ logic.
2. The idea is relatively novel, and alignment itself is a challenging problem.

**Weaknesses:**

1. The related work section misses several relevant studies (e.g., GeoReasoner [1] , Img2Loc [2]). I recommend that the authors do a more thorough survey.
2. In Table 2, which backbone is used for HierLoc? Is the comparison with other methods fair?
3. The code and data will only be released after acceptance.
4. Similar to the first point, the experimental section also lacks comparisons with strong baselines. The current results still show a gap from the state-of-the-art on some metrics.

Overall, I think this paper is well-presented, and the experiments are generally self-consistent. However, the lack of strong and up-to-date baselines is quite noticeable. If the authors can include comparisons with more recent baselines in this field, I would consider increasing my score.

[1] Li, Ling, et al. "Georeasoner: Geo-localization with reasoning in street views using a large vision-language model." Forty-first International Conference on Machine Learning. 2024.
[2] Zhou, Zhongliang, et al. "Img2Loc: Revisiting image geolocalization using multi-modality foundation models and image-based retrieval-augmented generation." Proceedings of the 47th international acm sigir conference on research and development in information retrieval. 2024.

**Questions:**

please refer to the weakness part

---

> ### Author Response · Authors · 2025-11-20
>
> Dear Reviewer,
>
> Thank you for recognizing the novelty of our work and thorough comments on the lack of missing comparisions.
>
>
> **Missing Related works and Baselines**
>
> The only reason we did not originally include the works you mentioned is that they rely on foundational LLM and VLM models. However, you are correct that these works should be part of the comparison. We have now included them in both the Related Works section and the experimental table as additional baselines. We have incorporated the officially reported results of GeoReasoner and Img2Loc into Table 2, since these methods evaluate on the same benchmarks and metrics. This allows for a fair comparison without requiring re-implementation. We have also discussed and report another work LocDiff as brought to our focus by reviwer V4da.
>
> **Table2 Backbone**
>
> Excuse us for missing this detail in the table description. It is based on DinoV3 backbone and as requested by reviewer k8zx we have further added more experiments about the effect of chosen backbone and to isolate the result gains to the HierLoc design and architecture rather than the choice of the backbone and reported these results in tables 4 & 5. We hope this helps about fairness concern.
>
> **Gap from SOTA**
>
> To clarify again we trained two models (one on OSV5M and one on MP16). we tested model trained on OSV5M with 210k OSV5M test images, the biggest open source test set available to evaluate global geolocation task. On this test set we achieve clear SOTA. The rest of the benchmark datasets, IM2GPS (297), IM2GPS3K(3000) and YFCC4K (4536) these test sets does not offer enough samples to statistically declare a SOTA model for this datasets and it is evident in the experiments table no model is a clear winner. At the same time our model achieves best or second best results for most of the metrics on IM2GPS, IM2GPS3K and YFCC4K. Thus the reasoning for us to claim SOTA only on OSV5M dataset.
>
> **Code Release**
>
> We are sorry about the situation with code release. Due to administrative constraints we have had difficulties with releasing our code at the moment, but we are working hard to do so in the future. We hope you understand.
>
> We appreciate your constructive feedback and believe the added comparisons and clarifications fully address the concerns you raised.

---

### Official Review · Reviewer_V4da · 2025-10-31

**Soundness:** 2
**Presentation:** 2
**Contribution:** 2
**Rating:** 2
**Confidence:** 4

**Summary:**

This paper is an incremental work on traditional retrieval-based geolocalization methods. Instead of performing contrastive learning between a query image and a full gallery of candidate locations, the authors propose to aggregate the candidate locations into a hierarchy of entities (country, region, subregion, city) and perform hierarchical contrastive learning in the hyperbolic space. This is an intuitive approach since it mimics the human spatial reasoning patterns -- narrow down from large-scale to fine-grained guesses. It shows some competitive performance against other baselines, while suffers from generalizability problems.

**Strengths:**

1. A hierarchical representation of locations and a beam search through hierarchical entities is a very natural and geographically sound alternative to traditional single-location-based retrieval geolocation methods.

2. Using hyperbolic embedding to represent hierarchy is very efficient.

3. Geo-Weighted InfoNCE introduces geo-awareness into the contrastive learning loss.

**Weaknesses:**

1. The key weakness, which severely restricts the generalizability of the method, is that the proposed entity hierarchy relies heavily on the **coverage** of the dataset. That is, if a location in the test set never appears in any neighborhoods of an entity in the training dataset, the model will never be able to predict its location. This is no rare case -- both MP16 and OSV5M are highly spatially biased, i.e. most data concentrate in North America and Western Europe. This is already a known problem in traditional retrieval based methods such as GeoCLIP (see a recent NeurIPS paper https://openreview.net/pdf/c2d943add9cd78700f9acc1101286c2082105a70.pdf, Section 5.2 and Appendix A.7), and the entity hierarchy which simplifies 4 million candidate locations into 240k entities only makes the problem worse. In other words, the proposed method, on its very basis, can not handle spatially out-of-distribution cases. This is a huge weak point compared to recently developed generative geolocation models (RFM, LocDiff) which naturally generalizes to arbitrary locations on Earth.

This problem is covered in the paper because the authors only performed testing on the same datasets the models are trained -- i.e., if a model is trained on the OSV5M training dataset, it is evaluated on the OSV5M test dataset. It "hides" the generalizability problem.

2. Errors in citations. Too many citations are in wrong formats -- e.g. throughout the paper, most citations are not properly put into parentheses.

3. From Appendix A.2, $\exp_0$ and $\log_0$ should be $\exp_o$ and $\log_o$, because here the $o$ represents the origin *point*, not the number $0$.

**Questions:**

1. Have you ever tested with cross-dataset experiments? For example, in GeoCLIP, the model trained on MP16 is evaluated on Im2GPS3k and YFCC, and obviously YFCC has much weaker performance because its data distribution is different from MP16. In your experiments, all test data seem to be a left-out of the training data. This will cover your generalizability issue, since the test locations are likely to align with the entity hierarchy you build from the training data.

---

> ### Author Response · Authors · 2025-11-20
>
> Dear Reviewer,
>
> We thank you for your constructive feedback. We value your concerns regarding generalizability and cross-dataset validation, and we have addressed them below with new experiments and clarifications.
>
> **1. Performance on Spatially OOD Scenarios (Generative vs. Retrieval)**
> You raised a concern that retrieval methods suffer in "data scarce" regions while generative models (like RFM) generalize naturally to arbitrary locations. To test this, we utilized the open-source code of **RFM** (the current SOTA generative model) to compare it directly against **HierLoc** on the "long tail" of OSV5M dataset.
>
> We isolated the **40 countries with the lowest number of training samples** in the OSV5M dataset (ranging from 5 to 639 samples). The results contradict the assumption that generative models handle these areas better:
> *   **HierLoc outperforms RFM in 28/40 (70%) of these data-scarce countries.**
> *   HierLoc achieves a lower Global Mean Error (**3878.7 km**) compared to RFM (**4440.6 km**) on this subset.
> *   **[Figure 1]** (linked below) visualizes the error difference. RFM frequently "hallucinates" in these regions (errors >7000km), while HierLoc’s hierarchical constraints maintain better bounds.
> *   **Evidence:** [Plot: HierLoc vs RFM on OOD Countries](https://0x0.st/KWuV.png) | [Raw Data (CSV)](https://0x0.st/KWuJ.csv)
>
> **2. Dependence on Coverage & Generalizability**
> We agree that HierLoc relies on dataset coverage; however, our analysis shows this limitation applies equally to generative models.
> *   **Visual Proof:** We generated a [Mean Error Distribution Map](https://0x0.st/KWut.gif) comparing HierLoc and RFM across 1-degree cells. Both models struggle in under-represented regions (outside North America/Europe). This confirms that "coverage dependence" is a dataset bias issue, not a flaw exclusive to retrieval methods. Note that OSV5M’s test split follows a strict 1 km spatial buffer from all training images. This results in many test areas that contain zero training samples within 1km. HierLoc must therefore generalize to previously unseen areas even within the same dataset, which is precisely what we analyze in the 40-lowest-sample countries.
>
> *   **Why HierLoc Generalizes:** Unlike flat retrieval (GeoCLIP), HierLoc mimics human reasoning. When a specific location is missing from the training set, the model falls back to the next best hierarchical entity (region/country) rather than making a random fine-grained guess. Although HierLoc cannot predict an unseen location to a fine-grained entity that does not appear in the training hierarchy, it can still produce a meaningful prediction by backing off to higher-level entities (region/country). This fallback mechanism is precisely what enables the strong performance we observe in completely under-represented countries
>
> **3. Cross-Dataset Validation (Table 2)**
> You asked if we tested cross-dataset experiments (e.g., MP16 $\to$ Im2GPS3k and YFFC4K).
> **We successfully performed these experiments in the original submission.** Please refer to **Table 2** (Main Paper), where we trained on **MP16 (4.8 M)** and evaluated on **Im2GPS3k** and **YFCC4k**.
> *   We agree with your observation that performance drops on YFCC4k due to distribution shifts.
> *   To contextualize this, we also evaluated RFM (trained on YFCC-48M) from [official implemention](https://github.com/nicolas-dufour/plonk) on these benchmarks and observed similar performance drops. This confirms the difficulty of the YFCC4k benchmark across all architectures including generative models. Please see rebuttal table 1.
>
> Rebuttal Table 1. RFM model trained on YFCC, tested on IM2GPS3K and YFCC4K
>
> | **Dataset** | **25km** | **200km** | **750km** | **2500km** |
> | ----------- | ---------| --------- | --------- | -----------|
> | Im2GPS3k | 29.2       | 43.9     | 62.6     | 80.1          |
> | YFCC4K   | 23.8       | 35.8     | 53.8     | 72.0          |
>
> **4. Comparison to LocDiff**
> We first would like to thank you for bringing this paper to our notice. The "uniform grid" generalization experiments in LocDiff are difficult to apply to HierLoc, which relies on administrative boundaries (entities) rather than arbitrary grid cells. However, our comparison above with **RFM** (the primary generative baseline) demonstrates that HierLoc remains competitive and often superior in regions with less training data. Nonetheless, it warrants a discussion about this paper and did so in the revised version.
>
> **5. Formatting & Citations**
> We have corrected the citation styles and the mathematical notation ($exp(0)$ vs $exp(o)$) in the final revision.
>
> We hope these additional experiments on spatially OOD regions and the clarification regarding Table 2 resolve your concerns about generalizability.

---

### Official Review · Reviewer_k8zX · 2025-10-31

**Soundness:** 4
**Presentation:** 3
**Contribution:** 3
**Rating:** 6
**Confidence:** 5

**Summary:**

This paper proposes to solve **geolocation as a hierarchical retrieval problem**. By leveraging different levels of administrative segmentation, they assign one embedding per entity and then perform **hierarchical retrieval to achieve efficient retrieval** while achieving **SOTA performance**.

**Strengths:**

1. The paper showcases **extensive experiments** to back up its claims.

2. The method's inference structure allows for **very efficient inference**.

3. The method is showcased on two datasets: **OSV-5M** (street view focused) and **MediaEval** (more generalist).

4. The authors achieve **SOTA performances on OSV-5M**.

**Weaknesses:**

1. **Data Curation:** It is not clear, but **entities seem to be learned *across* datasets** (per Section 3.2). If true, this **makes the results not comparable**, as the model is trained on significantly more data. This also potentially **breaks the data decontamination** for OSV-5M (1km exclusion zone). I would like to see **results with entities computed separately for each dataset**.

2. **Mean Embeddings for Large Regions:** Taking the mean embedding may be suitable for fine-grained regions, but it **likely doesn't make sense for wider regions** like countries. I would like to see an **ablation study sequentially removing these mean embeddings** for countries, regions, sub-regions, and cities.

3. **Importance of the Encoder:** The paper **fails to ablate the importance of the feature extractor**. For all we know, it’s **DinoV3 that is responsible for the improvement**. **Networks used by other SOTA methods should be ablated** as well (e.g., StreetCLIP on OSV, DinoV2 on YFCC for RFM S2).

4. **Lack of Qualitative Samples:** I would have loved to see some **visualisation of the inference process**. (Perhaps drawing the beam search trees on a map with colors for scores?) It would also beD; good to be able to visualise some final predictions.

5. **Efficiency Comparison:** I would like to see the **inference efficiency compared to non-retrieval methods** as well.

**Questions:**

**Motivation for Hyperbolic Embeddings:** The paper **struggles to clearly explain the intuition for using hyperbolic embeddings**. I understand that it works better, but what is the **underlying motivation**?

---

> ### Author Response · Authors · 2025-11-20
>
> Dear Reviewer,
>
> Thank you for the thoughtful, constructive review, and for recognizing the contributions of our work.
>
> **Q1. Data Curation**
> We apologize for the confusion caused by Section 3.2. We confirm that only the entity hierarchy is constructed across datasets, while the models themselves are trained fully separately:
> Model 1 (OSV5M): trained and evaluated on the official OSV5M train/test splits.
> Model 2 (MP16): trained on MediaEval16 (MP16) and evaluated on IM2GPS, IM2GPS3K, and YFCC4K.
> There is **no data contamination**, including adherence to the 1 km OSV5M buffer zone.
> To further address concerns, we additionally constructed entities only from OSV5M for the StreetCLIP ablation; following the same constructed entities only from MP16 for the MP16-trained model and report these results in Tables 4 & 5 in the revised version.
>
> **Q2. Mean Embeddings for Large Areas**
> Your intuition is correct: mean image embeddings contribute more strongly to finer-grained entities, while their effect diminishes for coarse-level regions. We therefore perform a sequential removal of mean embeddings (country → region → subregion → city). Importantly, the model does not collapse, even when mean embeddings are removed at every level. We have included this table in the revised paper as well (Table 10, Appendix A.5).
>
> Rebuttal Table 1. Mean Image Ablation Study
>
> | **Ablation (Removed Mean Embeddings)** | **Country (%)** | **Region (%)** | **Sub-region (%)** | **City (%)** |
> | -------------------------------------- | --------------- | -------------- | ------------------ | ------------ |
> | None                                   | 82.93           | 55.03          | 40.68              | 23.26        |
> | Country                                | 81.59           | 54.27          | 39.97              | 22.63        |
> | Country & Region                       | 79.96           | 50.40          | 36.61              | 20.59        |
> | Country & Region & Sub-region          | 77.12           | 44.34          | 30.76              | 16.32        |
> | Country & Region & Sub-region & City   | 70.46           | 30.00          | 14.06              | 3.58         |
>
> **Q3. Importance of the Encoder**
> We previously ablated the CLIP ViT-L/14 backbone in Table 1, but we agree that more encoder ablations strengthen the paper. We therefore include, StreetCLIP on OSV5M (entities constructed only from OSV5M) and DinoV2 on MP16 (entities constructed only from MP16) evaluated on YFCC4K. Table 4 from the revised paper confirms a bit better performance of StreetCLIP compared to CLIP (ViT-L14) backbone and confirms that the SOTA performance is not because of the choice of Encoder. Table 5 also shows that DinoV3 performs better than DinoV2 but still outperforms RFM $\mathrm{S_2}$ with 10x less training data. Given the character limit we have also [hosted this tables](https://0x0.st/KWuy.png) for easier access.
>
> **Q4. Qualitative Examples**
> Thank you for the suggestion. We will add qualitative examples of predicted locations, and a visualization of beam-search inference on a map, including node scores in the camera ready version given less time for rebuttal.
>
> **Q5. Inference Efficiency**
> To compare against non-retrieval methods, we evaluate HierLoc and RFM S2 on the same GPU.
>
> Rebuttal Table 2. Inference Latency Comparison per image
>
> | **Model**          | **Inference (ms)** | **Retrieval (ms)** | **Total Latency (ms)** | **Speedup vs RFM** |
> | ------------------ | ------------------ | ------------------ | ---------------------- | ------------------ |
> | **RFM S2**         | 14.17              | –                  | 14.17                  | 1×                 |
> | **HierLoc (ours)** | 2.09               | 0.075              | 2.16                   | **6.56× faster**   |
>
> **Motivation for Hyperbolic Embeddings**
> For the task of Visual geolocation exploiting the hierarchical structure is extensively researched in the works of OSV5M, PIGEON.
> Hierarchical structures (such as country → region → subregion → city) grow exponentially with depth. Hyperbolic spaces naturally model such exponentially branching trees. In Euclidean space at finer level, entities tend to collapse together due to limited representational capacity. Hyperbolic space expands volume exponentially with radius, enabling **clearer separation** between hierarchical levels, **larger representational capacity** for fine-grained entities, and **improved retrieval discrimination** during inference. This geometric property directly benefits our hierarchical retrieval pipeline. We have also added extra lines in the Introduction of the revised paper discussing the same.
>
> We thank the reviewer again for the insightful comments and suggestions, which helped us significantly strengthen the paper. We are happy to provide any additional clarifications if needed.

---

> > ### Comment · Reviewer_k8zX · 2025-11-20
> >
> > Thanks to the authors for the answer!
> > Could you highlight in red in the paper the parts that are changed so it's easy to parse?
> > Thanks!

---

> > > ### Author Response · Authors · 2025-11-20
> > >
> > > Dear Reviewer, Thank you for this suggestion. We have highlighted the changes in the paper for everyone to parse easily. Looking forward for your comments.

---

> ### Comment · Reviewer_k8zX · 2025-11-27
>
> I would like to thank the authors for their rebuttal.
> They clarified most of my points.
> The only one missing are qualitative examples and i trust the authors to have them in the camera ready.
>
> I think this paper is really high quality and shows very interesting speed and performance improvements to retireval based geolocation.
>
> I would very very interested given the results of the mean embeddings, to maybe try to figure out an hybrid method mixing generative approach like RFM and hierloc. The main shortcoming i see to Hierloc is that it lacks this generative aspect that allow users (journalist especially) to take informed decision given the predictions.
>
> I however think this paper is a very good contribution to the field and will therefore raise my grade to an 8

---

> > ### Author Response · Authors · 2025-11-28
> >
> > Dear Reviewer,
> >
> > Thank you very much for your thoughtful follow-up and for taking the time to carefully re-evaluate our work. We truly appreciate your detailed feedback throughout the discussion phase and are glad that the clarifications and additional experiments addressed your concerns.
> >
> > Your observations regarding the potential integration of generative components are especially valuable, and we agree that exploring hybrid approaches between RFM-style generative predictions and HierLoc’s hierarchical retrieval is a promising future direction. We will highlight this in the Discussion section of the camera-ready version.
> >
> > We also appreciate your trust regarding the qualitative visualizations. We will include them in the final paper.
> >
> > Thank you again for the constructive engagement and for improving the strength and clarity of our submission.

---

### Author Response · Authors · 2025-11-26

Dear Area Chairs, Senior Area Chairs, Program Chairs

As the review scores have been reverted to their pre-rebuttal state, we provide this concise summary. During the rebuttal period, we addressed all reviewer concerns with new experiments and clarifications, leading to a consensus on the novelty, technical validity of the work and a score increase from Reviewer k8zX.
_____________________________________________
**1. Critical Score Update (Reviewer k8zX: 6 → 8)**

After reviewing our rebuttal and additional experiments, Reviewer **k8zX raised their score to 8; accept, good paper (poster)**, to quote:

*“I would like to thank the authors for their rebuttal. They clarified most of my points… I think this paper is really high quality and shows very interesting speed and performance improvements… I will therefore raise my grade to an 8.”*

Due to global score reversion, this upgrade is **not reflected numerically**, but it is part of the open review record.
_____________________________________________
**2. Summary of Resolved Technical Concerns**

**Generalizability vs. Generative Models (Reviewer V4da)**
We compared HierLoc with RFM on the long-tail of OSV5M (the 40 lowest-coverage countries, 5–639 samples):

- **HierLoc outperforms RFM in 28/40 (70%)** countries.
- **Lower global mean error** vs. RFM.

This directly refutes the claim that hierarchical retrieval cannot handle spatially OOD regions.

**Data Fairness & Training Setup (Reviewers 7XjS & k8zX)**
We clarified that:

- Model A trains only on OSV5M (evaluated on OSV5M).
- Model B trains only on MP16 (evaluated on Im2GPS, Im2GPS3K, YFCC4K).

We added:
- Entity hierarchies built solely from OSV5M or MP16.
- Backbone ablations (CLIP, StreetCLIP, DINOv2, DINOv3).

These confirm improvements stem from the **HierLoc architecture**, not data volume.

**Missing / Strong Baselines (Reviewer ZDzY)**
We added the requested methods to Related Work and experiments. Reviewer ZDzY expressed **willingness to increase their score**, but did not provide a follow-up.

- Added GeoReasoner, Img2Loc, and LocDiff (official numbers).

Under these stronger baselines:
- HierLoc **retains SOTA** on OSV5M.
- Remains competitive on Im2GPS, Im2GPS3K, YFCC4K.

**Additional Improvements (Reviewers k8zX & 7XjS)**
- Mean embedding ablations across all hierarchy levels.
- Sensitivity for τ and λ (stable across wide ranges).
- Inference Efficiency comparison vs RFM S2: HierLoc is **≈6.5× faster** with negligible retrieval overhead.
______________________________________________
**3. Procedural & Field-Level Notes**

**Reviewer V4da (Score 2)**
Their rejection relies heavily on generalisation experiments from LocDiff, which is *not required* under ICLR policy:

- Authors need not compare to papers **published after July 24, 2025**.
- **LocDiff is accepted to NeurIPS 2025 on Sept 17, 2025**, after the cutoff.
- **LocDiff is not yet available as a peer-reviewed NeurIPS 2025 publication** at the ICLR deadline (only arXiv/OpenReview existed).

Thus, LocDiff is contemporaneous work; comparison to it cannot be grounds for rejection.
Regardless, we still discussed LocDiff and showed that HierLoc remains SOTA on OSV5M (including the LocDiff results), and outperforms generative methods such as RFM in low-coverage regions (LocDiff code is not public yet).

**Reviewer 7XjS (Score 4)**
They acknowledged our additional experiments and had no further technical questions, but maintained a low score due to **personal lack of interest**, writing:

*“I personally have limited interest in the research topic… the field is niche.”*

Reviewer **k8zX** responded:

*“Image geolocation is not niche, it is an active and growing field at NeurIPS and CVPR. Personal interest should not determine rejection decisions; evaluation should rely on scientific validity, novelty, and soundness.”*
______________________________
**Conclusion**

HierLoc achieves **state-of-the-art performance** on OSV5M and is **≈6.5× faster** than strong generative baselines. We provided targeted experiments addressing:

- Generalizability
- Fairness of training data
- Missing baselines
- Hyperparameter Sensitivity
- Architectural ablations

Given the resolved technical concerns and the upgraded recommendation from Reviewer k8zX (6 → 8), we respectfully ask that the AC consider the **full post-rebuttal record**. We would like to thank all the reviewers for taking time and raising important questions helping us substantially improve the quality of our work.

Sincerely,
The Authors.

---

### Public Comment · ~Hari_krishna_Gadi1 · 2026-02-27
**Camera ready Version**

Dear Program Chairs, Area Chair, and Reviewers,

We have uploaded the camera-ready version of HierLoc. We sincerely thank the committee for the positive recommendation and the insightful feedback provided throughout the review process.

To address the final requests from the Meta-Review and discussion phase, we have integrated the following updates:

**Geo-classification Context (Related Works)**: We have expanded our discussion to contrast our "hierarchy as representation" approach with traditional geo-classification methods (e.g., PlaNet, PIGEON), which primarily use hierarchy for spatial partitioning.

**Discretization Trade-offs (Discussion)**: We have added the discussion on the inherent 1km localization limit resulting from our 240k-entity discretization.

**Beam Search Visualization (Appendix)**: Per Reviewer k8zX’s request, we added Figure [5], which visualizes the hierarchical beam search process on a map, illustrating candidate refinement and associated node scores.

We are grateful for the guidance that helped strengthen the final manuscript and look forward to presenting our work at ICLR 2026.

Sincerely,

The Authors

---

### Meta-Review · Area_Chair_5kza · 2026-01-05

**Summary:**

The Area Chair recommends the acceptance of this paper. The work introduces a novel hierarchical entity-based approach to visual geolocation that demonstrates significant improvements in both speed (6.5× faster than generative baselines) and accuracy. During the rebuttal, the authors successfully addressed primary technical concerns regarding experimental fairness, data curation, and generalizability. Notably, they provided new experiments showing that HierLoc outperforms generative models like RFM in 70% of low-coverage "long-tail" countries, refuting the initial concern that the method cannot handle out-of-distribution regions. While the AC notes that the reliance on ~240k entities introduces an inherent discretization limit for extremely fine-grained localization (e.g., <1km) and suggests a deeper discussion on the parallels with geo-classification (considering geo-classes with different sizes), the work's overall technical validity is high. The AC expects that rebuttal would shift the reviewer consensus toward a positive recommendation, most notably with Reviewer k8zX increasing their score from a 6 to an 8. As the critical technical concerns seem resolved and the work establishes a new state-of-the-art on major benchmarks, the AC recommends acceptance.

**Reviewer Concerns:**

Resolved Concerns
- Generalizability to Low-Coverage Regions (Reviewer V4da): Initially, it was argued that an entity-based approach could not handle out-of-distribution (OOD) data. The authors' long-tail experiment on 40 low-coverage countries empirically refuted this, showing HierLoc outperformed generative models (RFM) in 28 out of 40 cases.
- Experimental Fairness and Data Volume (Reviewers 7XjS & k8zX): Concerns regarding training on combined datasets were resolved. The authors clarified that models were trained on isolated datasets (OSV5M-only and MP16-only) and provided backbone ablations (CLIP, StreetCLIP, DINOv2/v3) confirming the architecture, not just the data or encoder, drives the performance gains.
- Comparison to Recent Baselines (Reviewer ZDzY): The authors integrated comparisons with GeoReasoner, Img2Loc, and LocDiff. The results confirmed that HierLoc retains SOTA status on OSV5M even when compared to these more recent frameworks.
- Ablation Studies (Reviewers k8zX & 7XjS): Concerns regarding the importance of different hierarchy levels were addressed through mean embedding ablations across the entire hierarchy, and hyperparameter sensitivity was proved to be stable.
- Efficiency (Reviewer k8zX): The concern regarding inference overhead was resolved by demonstrating that HierLoc is roughly 6.5× faster than strong generative alternatives.

Outstanding/Minor Concerns
- Fine-Grained Localization Limits: While the rebuttal focused on regional accuracy, the AC believes the inherent discretization limit (due to the 240k fixed entities) for 1km localization remains a trade-off that should be more explicitly discussed in the final version.
- Contextualizing with Classification-Based Methods: The conceptual link to geo-classification with geo-classes at different sizes was not fully explored. The AC considers this an outstanding point for the related work section to properly position the paper.
- Personal Interest/Niche Field (Reviewer 7XjS): One reviewer maintained a low score based on a "limited interest in the topic." The AC agrees with the authors and Reviewer k8zX that scientific validity and performance should take precedence over a reviewer's personal assessment of a field's "niche" status.

**Reviewer Scores:**

Reviewer k8zX: Specifically raised the score from 6 to 8 (reverted but mentioned in the comment)

Reviewer ZDzY: Requested discussions and experiments with additional baselines and the authors provided them addressing the concerns. The reviewer mentioned that they will raise the score once this question is properly answered as the paper is well-presented.

Reviewer V4da: Core technical objection (OOD failure) was empirically refuted. Though likely still preferring generative models, the primary grounds for rejection were invalidated.

Reviewer 7XjS: Score remains stagnant due to personal lack of interest in the topic ("niche field"), despite acknowledging the success of the new experiments.

---

### Decision · Program_Chairs · 2026-01-26

Accept (Poster)